# Accelerated lysine metabolism conveys kidney protection in salt-sensitive hypertension

Markus M. Rinschen [1,2,3,4,20✉], Oleg Palygin [5,20], Ashraf El-Meanawy [6], Xavier Domingo-Almenara [1,7], Amelia Palermo[1,19], Lashodya V. Dissanayake [8,9], Daria Golosova [9], Michael A. Schafroth [10], Carlos Guijas[1], Fatih Demir [2], Johannes Jaegers [2], Megan L. Gliozzi[11], Jingchuan Xue[1], Martin Hoehne [12,13,14], Thomas Benzing [12,13,14], Bernard P. Kok[15], Enrique Saez [15], Markus Bleich [16], Nina Himmerkus[16], Ora A. Weisz[11], Benjamin F. Cravatt [10], Marcus Krüger [12,13], H. Paul Benton [1], Gary Siuzdak [1,20✉] & Alexander Staruschenko [8,9,17,18,20✉]

Hypertension and kidney disease have been repeatedly associated with genomic variants and alterations of lysine metabolism. Here, we combined stable isotope labeling with untargeted metabolomics to investigate lysine's metabolic fate in vivo. Dietary $^{13}C_6$ labeled lysine was tracked to lysine metabolites across various organs. Globally, lysine reacts rapidly with molecules of the central carbon metabolism, but incorporates slowly into proteins and acylcarnitines. Lysine metabolism is accelerated in a rat model of hypertension and kidney damage, chiefly through N-*alpha*-mediated degradation. Lysine administration diminished development of hypertension and kidney injury. Protective mechanisms include diuresis, further acceleration of lysine conjugate formation, and inhibition of tubular albumin uptake. Lysine also conjugates with malonyl-CoA to form a novel metabolite Nε-malonyl-lysine to deplete malonyl-CoA from fatty acid synthesis. Through conjugate formation and excretion as fructoselysine, saccharopine, and Nε-acetyllysine, lysine lead to depletion of central carbon metabolites from the organism and kidney. Consistently, lysine administration to patients at risk for hypertension and kidney disease inhibited tubular albumin uptake, increased lysine conjugate formation, and reduced tricarboxylic acid (TCA) cycle metabolites, compared to kidney-healthy volunteers. In conclusion, lysine isotope tracing mapped an accelerated metabolism in hypertension, and lysine administration could protect kidneys in hypertensive kidney disease.

A list of author affiliations appears at the end of the paper.

In the past decade, the role of amino acid metabolism has extended from a provider of protein building blocks to a decisive element in various physiological processes and diseases[1,2]. Lysine is an essential amino acid at the crossroads between proteins and metabolism and is of interest as both drug target and modifier of protein activity[3–5]. Lysine incorporated into proteins is heavily modified with increasing numbers of molecules such as organic carboxylic acids[6–10]. Lysine from protein residues also gives rise to carnitines with a central role in beta-oxidation. Multiple molecular lysine degradation and modification reactions are known, but its related bioactivity[11] has not been studied in a systematic fashion. Untargeted metabolomics offers the ability to identify altered and previously undescribed physiologically relevant molecules[11]. When applied together with isotope-labeling (using $^{13}C$ labeled molecules), the kinetics of synthesis and degradation can be quantified, and previously undescribed metabolites can be discovered.

The kidney is a central hub of mammalian metabolism, but its metabolic importance is relatively understudied. With up to one out of ten persons suffering from kidney disease, kidney diseases pose a significant socioeconomic burden[12]. Hypertension, potentially caused by kidney disease, is also an epidemic, and both entities are strongly associated[13]. Multiple genome-wide association studies have highlighted the role of carnitine metabolism and lysine transport (via kidney-specific transporter SLC7A9)[14–16] in kidney disease. Metabolomic biomarker studies in the Framingham cohort, one of the most important cohorts for predicting cardiovascular outcomes, showed that a high abundance of lysine in the urine is protective[17]. Three randomized controlled trials (RCTs) in humans performed untargeted metabolomics of serum and urine under dietary conditions beneficial for hypertension (so-called DASH diet, or sodium restriction)[18–20], and found an association with high signals of lysine, or modified lysine, with protective interventions.

The aim of this paper was to establish an untargeted metabolomics workflow to interrogate ten body organs of a mouse that were fed with an isotope-labeled diet $^{13}C_6$ lysine chow for up to two months and to utilize this knowledge to intervene with hypertensive kidney disease. The analysis revealed that the kidney is strongly involved in handling lysine metabolites, with a specific bias towards conjugating free lysine to modifications at a faster speed. Notably, the identity of lysine-conjugated molecules formed, and lysine's metabolic activity can be exploited to induce intense metabolic alterations driven by excess lysine to orchestrate a kidney-protective effect in hypertension.

## Results

**Isotope tracing of dietary $^{13}C_6$ lysine.** Adult male mice were fed with a normal diet and were switched to food containing exclusively $^{13}C_6$ lysine for 0 (control), 1, 2, and 8 weeks (Fig. 1A). Unlabeled mice served as a control. A significant shift in global metabolomics features occurred depending on the duration of the diet (Fig. S1A). The analysis consisted of two approaches: (1) a mass-difference approach that looks for coeluting peaks (modified from the previous protocols[21]), and (2) a correlation-based approach that extracts the mass trace of provided compounds and uses correlations to call out isotopologues. The analysis revealed 66 metabolites (approach 1) as well as 133 metabolites (approach 2) that were validated using standards and fragmentation MS2 spectra (Fig. 1B). Both methods found a significant overlap of validated metabolites. Overall, the correlation of extracted and verified metabolites was higher than other non-verified ones (Fig. S1B, C). Analysis of isotopologue patterns revealed that virtually all these metabolites were labeled with $^{13}C_6$ or $^{13}C_4$ atoms, meaning their entire carbon backbone was incorporated (Fig. 1C). Rates of $^{13}C_4$ incorporations were

significantly slower as compared to $^{13}C6$ units (Fig. S1D). We then separated the labeled peaks into different classes of lysine metabolites that included lysine, lysine degradation products such as saccaropine, pipecolic acid, and aminoadipic acid, modified lysines such as N- and E-acetylation, methylation, hydroxylation and many others, carnitine synthesis products such as trimethyllysine, lysine-containing peptides, and acylcarnitines. Detected lysine metabolites, as well as chemical information, are listed in Supplementary Data 1 and 2. In general, we found using both methods that lysine had the highest incorporation slope among the metabolites, followed by lysine degradation and modification and by peptides. In general, the incorporation of lysine into acylcarnitines was slower than other metabolic processes (Fig. S1D). We calculated the slope of different classes of metabolites using a linear model during the first two weeks. Notably, we did not observe any labeling in $^{13}C_2$ units (such as glycine), and targeted analysis of the TCA cycle metabolites revealed no significant incorporation of TCA metabolites in the TCA cycle in any tissue (summary in Fig. S2). Analysis of the diet using untargeted metabolomics revealed that the food had no detectable amount of carnitines, and other lysine metabolites in the diet (Fig. S1). A comparison of different organs was performed next, and slopes of incorporation were summed and plotted. Organ-specific incorporation rates (plotted as the slope of difference of leading $C_{13}/C_{12}$ isotope) of lysine metabolite classes were distributed to the different organs (Fig. 1D). In the thymus, we observed several lysine-incorporated peptides that were represented in this organ. The analysis revealed that the kidney cortex was one of the fastest incorporating organs, especially for modified lysines, peptides, and acylcarnitines (Fig. 1D). The kidney and liver showed the highest incorporation rates, but the kidney had a higher incorporation rate for Ne-Fructosyl-lysine (Fig. 1E).

In an independent experiment, we asked whether metabolome or proteome were leading in terms of lysine modifications. Kidney cortex tissue from mice fed with $^{13}C_6$ lysine for 1, 2, and 3 weeks was analyzed. Since lysine also gets incorporated into proteins, we determined both the metabolome and the proteome from these samples. We plotted the logarithmic ratios of the maximum isotopologue (peptides labeled with a mass shift of 6 Da) over control (Fig. 1F). The isotopologue ratio for lysine was higher than any other molecule, followed by metabolites from the lysine degradation pathway and modified lysines. Proteins were incorporated significantly slower as compared to lysine metabolite, and histone and DNA-binding proteins had the lowest incorporation rate among proteins (Fig. 1F). Metabolites ratio of labeled over unlabeled molecules had a wider distribution compared to proteins (Fig. 1G). To further investigate if protein incorporation is different in the medulla, we micro-dissected the kidneys in four macroscopic areas: outer stripe cortex, outer stripe medulla, inner stripe outer medulla, and inner medulla (Fig. S3A). We found that the lysine protein incorporation was slower in the medullary areas, consistent with the known metabolic role of the proximal tubule (Fig. S3B). Taken together, these data suggest that the cortex, and specifically the proximal tubules that comprise more than 80% of the cortex, is likely to be the major hub within the kidney for lysine metabolism.

**Lysine metabolism is accelerated in the Dahl salt-sensitive rat.** We had earlier reported a deficit of lysine metabolites in the kidney cortex in a hypertensive kidney disease model[22], the Dahl salt-sensitive rat. Dahl salt-sensitive (D/SS) rats are a physiologically relevant model of hypertension and hypertensive kidney disease that generates high blood pressure and kidney damage when fed a high salt diet. Based on these findings and the potential beneficial role of lysine in the Framingham cohort[17] and three randomized controlled dietary intervention trials[18–20], we

hypothesized that hypertensive kidney disease alters lysine metabolism. To understand whether lysine metabolism is altered in hypertension, we performed in vivo isotope labeling of 15 Nε-lysine. This strategy allows us to distinguish the relative contribution of Nε degradation (to the pipecolic acid pathway) as well as to the amino adipic pathway (canonically via the Nα amino group) (Fig. 2A). We injected hypertensive and non-hypertensive rats with 15 Nε lysine i.p. and analyzed serum and organ tissue for incorporated lysine metabolites (Fig. 2B). Figure 2C summarizes the quantified metabolites, as well as their incorporation in different tissues when animals were treated with normal or high salt diets. Hypertension markedly accelerated lysine metabolism towards lysine modifications in the kidney and liver (Fig. 2D). To investigate the direct metabolic kidney capabilities, we incubated kidney proximal tubule suspensions with 15 Nε lysine. We found that tubules are able to transfer lysine to saccharopine rapidly but are not capable of transforming lysine into pipecolate which was not detectable (Fig. 2E). Increased formation of labeled aminoadipic acid was not detected. Instead, non-labeled amino-adipic acid (AAA) increased time-dependently in the tubules, suggesting high and direct capacities of the kidney for Nα-mediated degradation (Fig. 2E).

**High lysine administration protects kidneys in hypertension.** Having clarified the mechanism of lysine degradation in the kidney, we hypothesized that the administration of lysine could be exploited in a model of hypertensive kidney disease. We subjected hypertensive rats to a high abundance of lysine in the drinking water (Fig. 3A). We monitored real-time blood pressure using telemetry over the entire time course of the disease (Fig. 3B). Both male and female rats on different salt diet regimens receiving lysine supplementation were largely protected with attenuated development of hypertension (Figs. 3B, S3A). Albuminuria, a readout of hypertensive kidney damage, was also decreased (Figs. 3C, S3B). The excretion of potassium, sodium, calcium, and chloride did not change between lysine-treated and non-treated rats, suggesting that the hypertension-inducing salt diet was not taken up to a different extent (Fig. S3C–F). Automated histology analysis using convoluted neural nets[23] revealed a significant improvement in the ultrastructure of glomeruli by lysine (based on quantification of more than 3500 glomeruli/group) (Fig. S3G). Functional analyses of hormones regulating blood pressure were unchanged or showed increased activity with lysine treatment, excluding a central regulation for propagating lysine's protective effect (Table 1).

To understand lysine handling in high and low lysine conditions, we performed in vivo isotope labeling of [15]N labeled lysine in lysine-treated and non-treated hypertensive rats. We injected lysine-treated and lysine-untreated rats with a defined bolus of [15]N lysine (Fig. 3D) and analyzed urine (collected 24 h). Less labeled lysine was detected in the urine of ad libitum lysine-treated rats (Fig. 3E). We also found an increased abundance of labeled saccharopine, Ne-acetyllysine, and fructoselysine in urine (normalized by creatinine) (Fig. 3E). Consistent with an increased lysine clearance, physiological studies measuring 24 h urine revealed that lysine induced diuresis (Fig. 3F), a finding observed after only 1 day of lysine treatment (Fig. S3H).

**Physiological effect of lysine on proximal tubule function.** Hypertension results in proteinuria, and proteinuria is further detrimental for kidney function when urinary proteins, albumin mainly, are taken up by the proximal tubule, the predominant cell type within the kidney cortex[24]. Therefore, we hypothesized that albumin uptake in the proximal tubule was blocked or altered by lysine, supported by previous in vitro and in vivo findings[25] in a physiological context. Consistent with this hypothesis, lysine

administration itself in D/SS rats on a normal salt diet induced an increase in urinary albumin (Fig. 4A). Cell culture studies using Opossum Kidney (OK) proximal tubule cells revealed that even moderate addition of lysine decreased uptake of fluorophore-labeled albumin (Fig. 4B), a key function of proximal tubules. Basolaterally-applied lysine (2 h, 50 mM) inhibited uptake of albumin, even when the cells were assayed 2 h after lysine washout (Fig. S4A). This is remarkable because the receptor LRP2 responsible for albumin uptake is localized apically[26]. In addition, we observed that overnight incubation with lower doses of lysine resulted in a profound and dose-dependent decrease in albumin uptake (half max inhibition at ~3.5 mM lysine) (Fig. S4B). Next, we examined hypertensive kidneys after dietary high-salt exposure in vivo using dual photon microscopy and fluorescently labeled albumin (Fig. 4C). Utilizing the injection of labeled fluorescent markers, the method allows for in vivo insights into albumin handling. Proximal tubules took up albumin and accumulated excessive albumin in luminal protein casts (arrows), suggesting that overloading of the albumin uptake machinery occurs in hypertension. Consistently, lysine reduced albumin casts in kidneys as observed in Masson's trichrome staining (Fig. 4D). The distribution of LRP2, the key uptake mechanism of proteins, changed in hypertension pathology from a uniform distribution to a patchy pattern with areas of complete absence of reabsorption in dilated tubules[25]. This effect was completely diminished by lysine administration. Expression of kidney injury molecule-1 (KIM-1), a marker of proximal tubule injury, was strongly decreased with lysine as well, suggesting protection from tubule damage by lysine (Fig. 4D).

Combined, in vivo and in vitro observations argue for an effect of lysine beyond competitive inhibition of albumin binding to the apically-localized megalin and cubilin receptors[26]. To further investigate the mechanism, we examined the cortex metabolome (Fig. 4E). Kidneys from healthy rats (on a low-salt diet) did not change lysine or kidney metabolism at all when challenged with a high lysine intervention ($q < 0.05$). However, with increased disease severity, there were more alterations in lysine metabolites and general metabolism. Lysine was increased in hypertension, as also supported by HPLC analysis (Fig. S4C). We found that especially metabolites that had faster incorporation in the initial experiment (such as Ne-acetyllysine, saccharopine, and fructose-lysine), were increased through the lysine challenge, suggesting these metabolites serve as an "overflow" (Fig. 4E).

To further investigate the changes induced by lysine on the kidney tissue, we performed proteomic analysis. Proteome analysis of the kidneys revealed an increase in the metabolic enzyme AASS; an enzyme that produces saccaropine by lysine oxidation and reaction with alpha ketoglutarate (Fig. S5A), and leads to a release of $NAD^+$. This regulation may explain the substantial increase in the saccaropine. Interestingly, there were no significant alterations in the kidneys of the key salt transporters, including NKCC2, NCC, and SGLT2 (Fig. S5B). Data are available in Supplementary Data 4.

Further analysis of the metabolomics dataset suggested that free fatty acids (FFA) were altered by lysine supplementation (Fig. 4F), with a strong decrease in the most severe damage condition. The entire class of FFA ($n = 22$ quantified molecules) showed a distinct and robust downregulation, with linoleic acid being decreased by lysine treatment in every condition. Further examination of the dataset revealed a reduction of glucose and other sugar metabolites. $NAD^+$, a kidney protective molecule associated with decreased oxidative stress, was increased with lysine treatment in severe conditions (Fig. 4G), and AMP was distinctly regulated in the kidney through lysine (increased at a high salt diet with 2 weeks, and decreased at high salt diet for 3 weeks), (Supplementary Data 3).

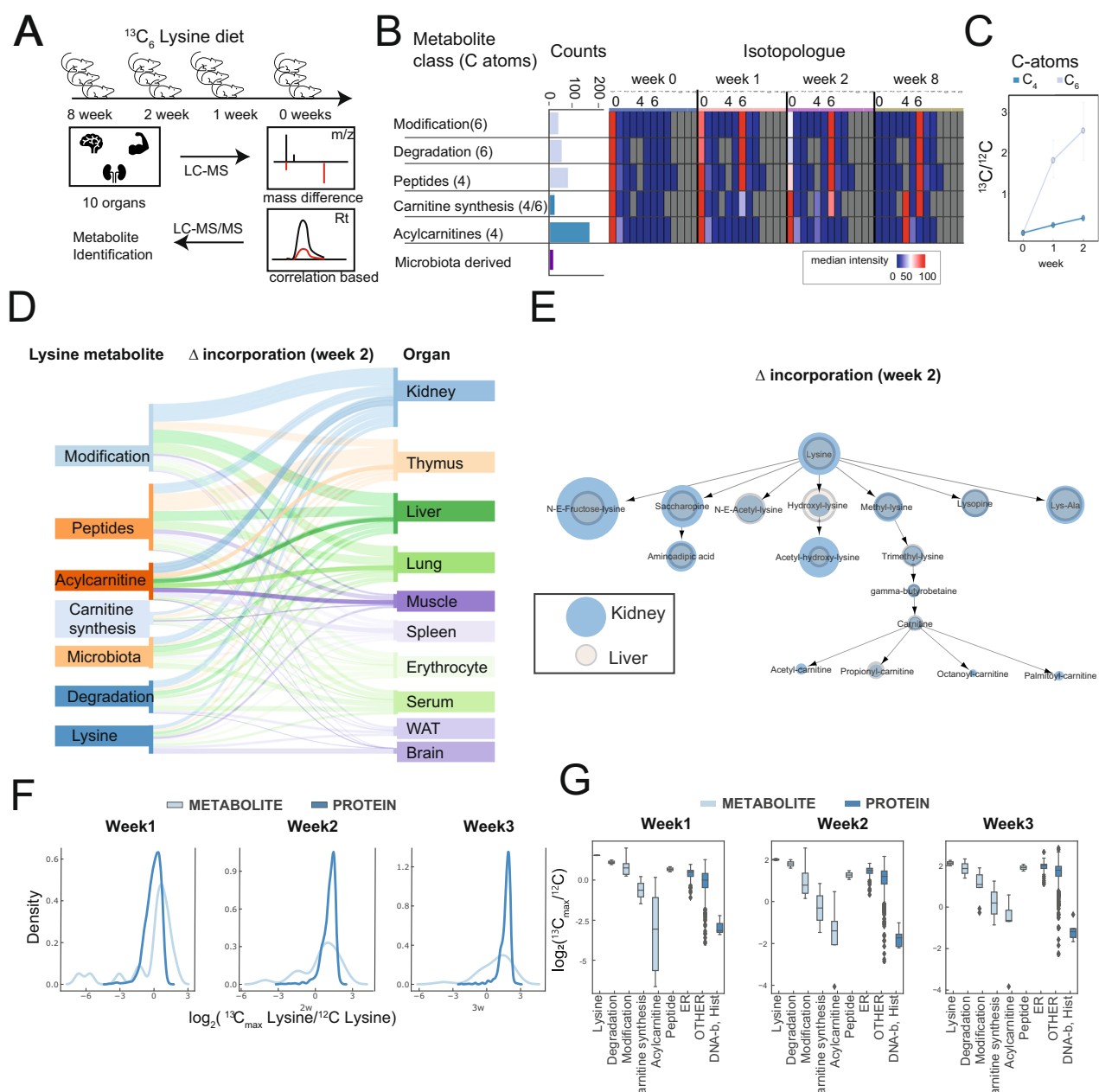

**Fig. 1 Untargeted analysis of the lysine-labeled metabolome reveals organ-specific lysine handling. A** Mice were labeled in triplicates with food containing exclusively $^{13}C_6$ Lysine for 1, 2, and 8 weeks. **B** Overview of main labeled metabolite classes, and their combined isotopologue patterns over time. **C** Slope of isotope incorporation of lysine metabolites with $C_4$ vs $C_6$ backbone (summarized metabolites from $n = 3$ isotope-labeled mice), error bar = SD. **D** Sankey diagram summarizing incorporation slopes of different classes of lysine metabolites in different organs. **E** Comparison of slopes of incorporations in liver vs kidney. **F** Comparison of lysine incorporation into the metabolome and the proteome of kidney cortex. **G** Density of lysine incorporation into the metabolome and proteome of the kidney cortex (summarized metabolites from $n = 4$ mice per time point). Dark blue lines are protein, light blue lines are metabolite density. Source data for this figure is available.

**Malonyl-lysine links protein modification and free fatty acid metabolism.** We hypothesized that lysine's tendency to form conjugates might contribute to the reduction in the FFAs. We examined the isotope labeling dataset for lysine conjugates that could be responsible for a decrease in fatty acid building blocks. Careful examination of the unknown peaks in the untargeted metabolomics isotope study revealed the presence of a mass consistent with Nε-malonyl-lysine in the kidney and liver (Nε-malonyl-lysine, $m/z$ [H+] of 233.11 and $m/z$ as a Na adduct of 255.10) (Fig. 5A). This molecule is a previously undescribed metabolite without a Pubchem entry. Synthesis of a standard,

together with retention time comparison (Fig. S6A, B, Supplementary material and methods), confirmed the identity of the compound as Nε-malonyl-lysine (Fig. 5B), and targeted analysis using MRM showed labeling only the $^{13}C_6$ lysine containing kidney tissue (Fig. S6C). In order to understand if malonyl-lysine would form enzymatically or chemically, we analyzed whether it would form in alkaline conditions from the educts malonyl-CoA and lysine (10:1 w/w) ratio. We observed that <1% of lysine molecules were malonylated (Fig. 5B). We then checked whether formation of malonyl-Lysine was also accelerated in the presence of hypertension and lysine treatment. Using bolus-labeled isotope

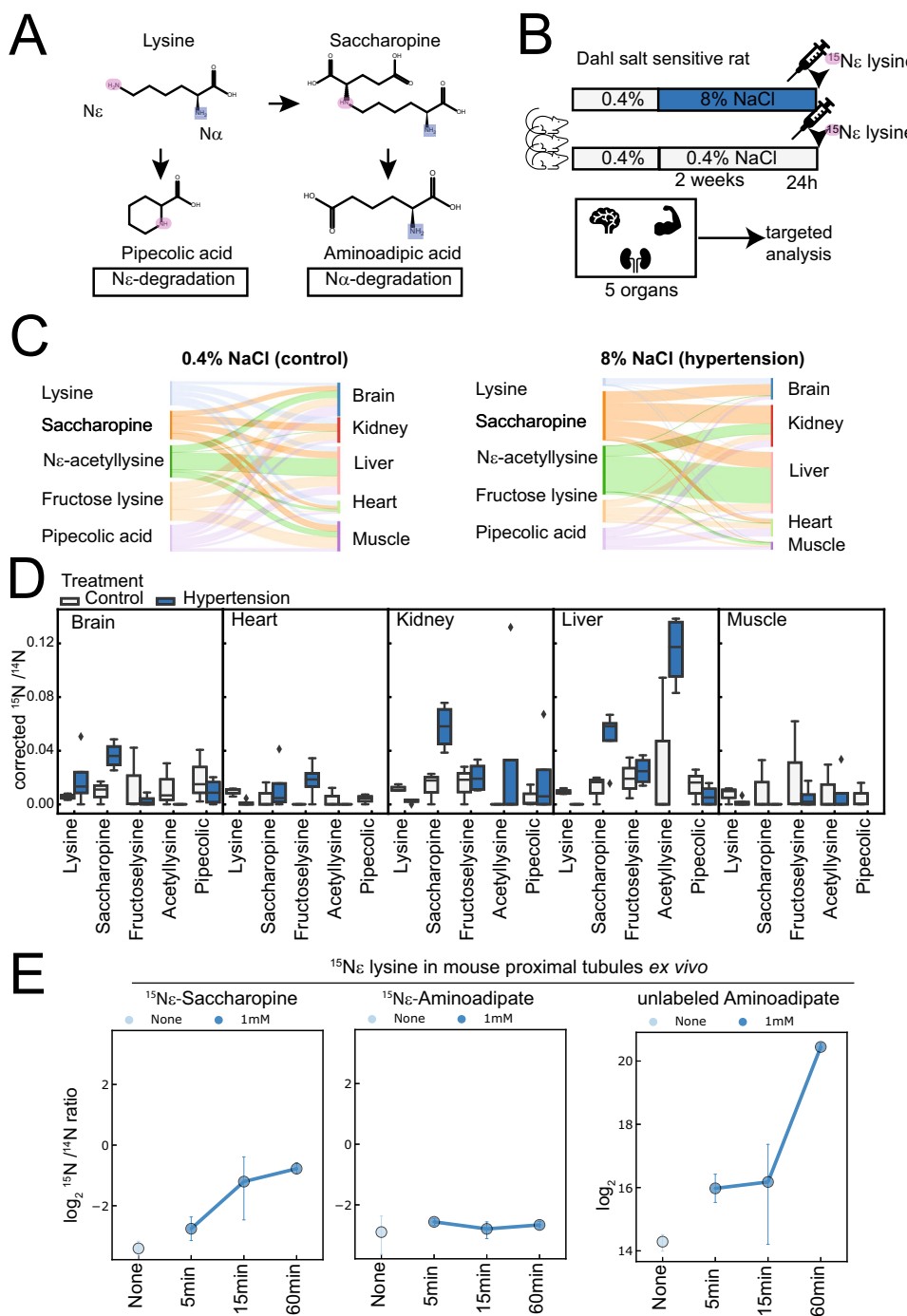

**Fig. 2 Targeted analysis of the lysine metabolism reveals renal alterations in hypertension and degradation via the Nα-saccharopine pathway.**
**A** Chemical pathways for lysine Nα- and Nε-degradation. **B** Experimental design to analyze lysine fate in Dahl salt-sensitive rats. **C** Overview of lysine metabolite isotope signal in **D**. Corrected ratios of $^{15}N/^{14}N$ labeled compounds in different organs ($n = 3$ rats/group). **E** Proximal mouse tubules were incubated with $^{15}N\varepsilon$ lysine ex vivo ($n = 3$ independent experiments performed in time course), error bar = SD. Source data for this figure is available.

tissue, we found that lysine administration increased and accelerated the metabolization of 15Nε-labeled lysine to 15 Ne-labeled malonyl-lysine (Fig. 5C). We found that Nε-malonyl-lysine was significantly increased in both hypertensive kidneys and controls with lysine (Fig. 5D), suggesting that lysine supplementation could deplete the abundance of malonyl-CoA, a substrate of fatty acid synthesis and an inhibitor of fatty acids. Consistently, we observed a significant decrease of malonyl-CoA (Fig. 5E). In addition, Nε-acetyl-Lysine was increased by lysine treatment (Fig. 5F). Acetyl-CoA was decreased in at least one lysine

treatment condition and trended to be reduced in the other two conditions (Fig. 5G). While the free acetyl-lysine and Nε-malonyl-lysine were increased, protein modification through these metabolites decreased, as evidenced by immunoblotting (Fig. 5H). These findings suggest that the chemical modification of lysine occurs in the kidney to deplete important anabolic molecules Acetyl-CoA and malonyl-CoA.

**Lysine reacts with central carbon molecules to excrete them.**
We then analyzed the metabolic changes more integratively,

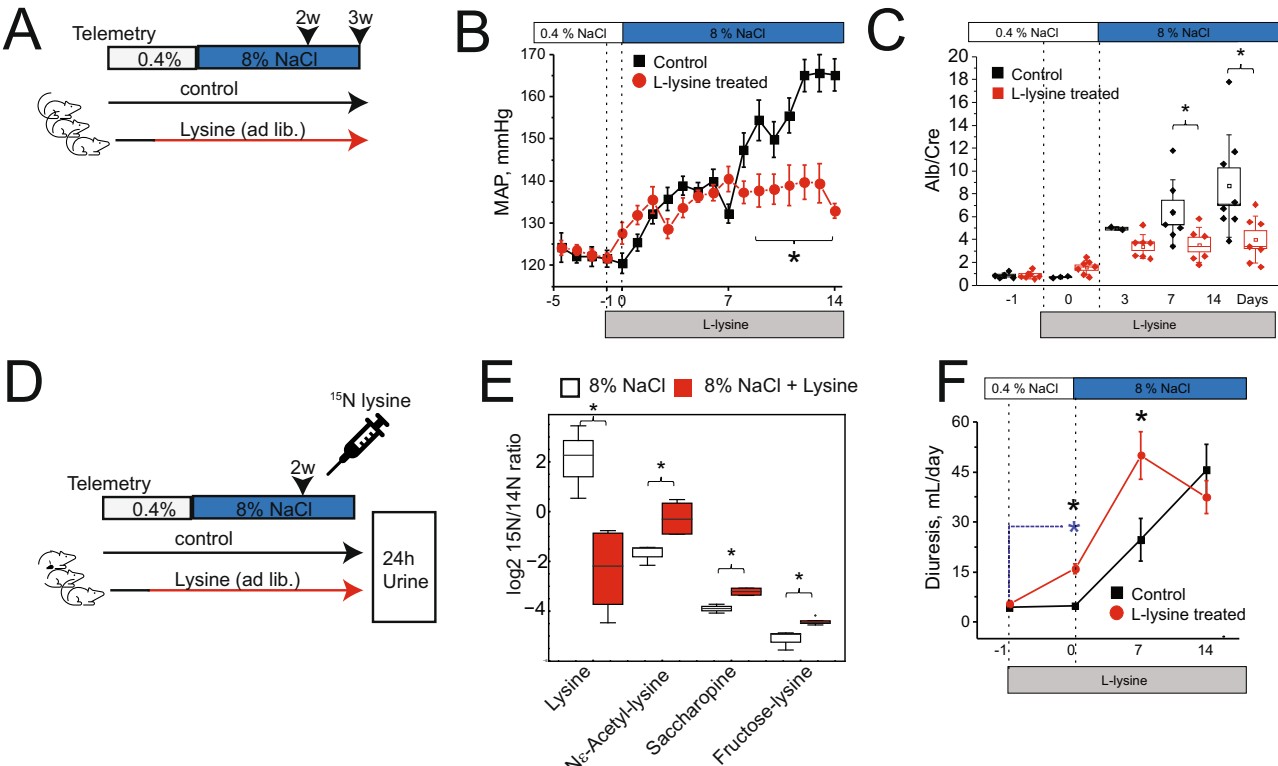

**Fig. 3 Counteracting lysine deficiency in hypertension ameliorates the disease by physiological mechanisms. A** Lysine administration protocol for rats. **B** Mean arterial pressure (MAP) in male D/SS rats under L-lysine treatment ($n \geq 7$ animals per group), error bar = SEM. **C** Albuminuria (normalized to creatinine) in control and L-lysine treated D/SS rats ($n = 6$ animals per group), error bar = SD. **D** $^{15}$N-lysine administration and lysine treatment protocol for rats. $^{15}$N isotope-labeled lysine treatment protocol to analyze lysine trace in presence of excess lysine. **E** Metabolomic analysis and signal distribution of $^{15}$N lysine-labeled metabolites in urine ($n = 3$ animals). **F** Lysine amplifies hypertension-induced diuresis (diuresis/24 h) ($n = 7,8$ animals), error bar = SEM. Source data for this figure is available.

**Table 1 Attoquant-based analysis of the Renin-Angiotensin Aldosterone system (RAAS) components of rats on a high salt diet with and without lysine.**

| RAAS component | SS-HS (pmol/l) | SS-HS + lysine (pmol/l) |
|---|---|---|
| Angiotensin II (1–8) | 204 ± 37 (N = 9) | 471 ± 136 (N = 8) |
| Angiotensin 1-7 | 10 ± 2 (N = 9) | 29 ± 9 (N = 8)* |
| Angiotensin I (1–10) | 214 ± 45 (N = 9) | 531 ± 134 (N = 8) |
| Angiotensin III (2–8) | 11 ± 3 (N = 9) | 18 ± 6 (N = 8) |
| Angiotensin 1-5 | 11 ± 3 (N = 9) | 37 ± 9 (N = 8)* |
| Angiotensin IV (3–8) | 11 ± 3 (N = 9) | 21 ± 6 (N = 8) |
| PRA (plasma renin activity) (Ang I + Ang II) | 418 ± 81 (N = 9) | 1002 ± 268 (N = 8)* |
| ACE activity (Ang II/Ang I) (no units) | 0.99 ± 0.04 (N = 9) | 0.89 ± 0.06 (N = 8) |

evaluating both matched tissue, urine, and serum from the same animals and calculating correlation coefficients between individual metabolites in different compartments. Integrative analysis of correlations between tissue and urine metabolites revealed that urinary lysine was strongly associated with tissue lysine and tissue lysine degradation products (Fig. 6A). High lysine in the urine was also negatively correlated with proteinuria (Fig. 6B), consistent with the observed long-term protective effect in the animals (Fig. 3). Interestingly, urinary lysine was also negatively correlated with the abundance of sugar molecules, such as glucose, glucose-6-phosphate, and sucrose. In fact, the targeted

analysis revealed a decrease in glucose abundance in the cortex. Together with an increase of Nε-fructoselysine in the urine, this suggests a net glucosuric effect. Interestingly, this effect was stronger in severely damaged cortices. Lysine treatment also induced loss of other amino acids under these conditions, consistent with previous reports[22] (Fig. 6C). The strongest excretion of lysine was as a sugar conjugate (fructoselysine), acetyl-CoA (acetyl-lysine), and α-ketoglutarate (saccharopine) in the damaged conditions (Fig. 6C). Analysis of lysine stoichiometry (mol/per ug creatinine) revealed that most of the excreted lysine was in the unmetabolized form. However, the excretion of lysine in the form of saccharopine, fructoselysine was more significant when kidneys were protected by lysine (Fig. 6D). The tissue abundance of the final metabolites of the TCA cycle, fumarate, and malate, were significantly depleted by lysine (Fig. 6E). Metabolomic analysis of serum revealed that in hypertension, there were no major changes in fructose lysine, saccharopine, acetyl-lysine, and others (Fig. S7), as opposed to control rats. Taken together, these results suggest that excessive lysine reacts with active metabolites of the central carbon mechanism to excrete them, suggesting metabolic alterations may play a role in the beneficial effects of lysine in this model.

**Lysine has no effect on isolated hypertension**. The obtained data does not show whether the observed metabolic changes are typical for kidney disease, hypertension, or both. To further clarify the mechanism, we subjected spontaneously hypertensive rats (SHR) to the same lysine intervention (Fig. 7). The SHR rat is a well-established model of hypertension[27]. In contrast to Dahl and other salt-sensitive rat strains, the SHR develops only

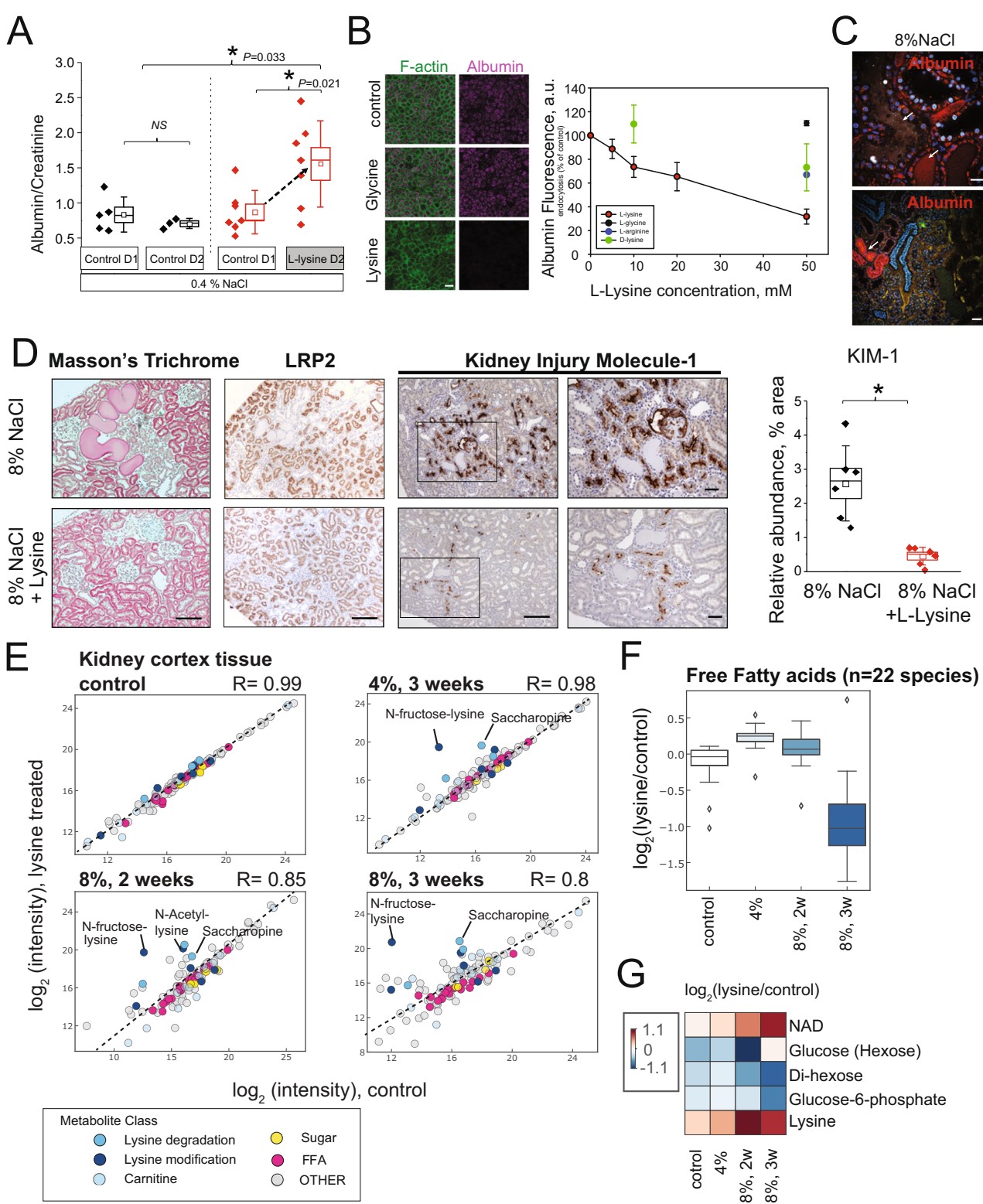

minimal kidney damage after salt supplementation[28]. Feeding experiments did not reveal a protective effect on hypertension, as well as no short- or long-term effects on albuminuria (Fig. 7A, B). Further targeted analyses revealed no renal transformation in lysine modification metabolites using an abundance of Nε-acetyllysine as a marker (Fig. 7C), as directly compared to the DSS rat model. We then asked if key metabolite changes were different in SHR and DSS rats. We analyzed TCA metabolites. Indeed, the DSS model reduced the abundance of urinary TCA

metabolites, but the SHR did not (Fig. 7D). This argues for an overall significant perturbation of metabolism in the DSS rat but not in the SHR rat that does not develop kidney damage.

**Acute lysine modulates proximal tubule function in humans.** We, therefore, suggest that accelerated lysine metabolism can be leveraged in hypertension associated with kidney damage. We performed a pilot human physiology study that exposed patients

**Fig. 4 Lysine treatment directly targets the proximal tubule. A** Urinary albumin/creatinine ratio of short-term (24 h) lysine treated D/SS rats on a normal salt diet ($n \geq 5$ animals per group). **B** Lysine blocks albumin uptake in the proximal tubule cells (OK cell culture) (right panel; green F-actin, magenta fluorescent-labeled albumin). Dose-response showed the inhibition of proximal tubule cell albumin uptake by different lysine concentrations ($n \geq 3$ independent cell cultures per group), error bar = SD. **C** In vivo imaging of protein casts and albumin endocytosis by dual photon microscopy. Albumin is labeled red and both protein plaques and failed endocytosis can be visualized in hypertensive D/SS rats. **D** Lysine supplement prevents proximal tubule injury in D/SS rats on a high salt diet. Reduction of tissue damage and protein plugs with lysine. Substantial reduction in megalin abundance (LRP2) in dilated proximal tubules can be restored by lysine supplementation. The presence of KIM-1 on the proximal tubule apical membrane shows severe kidney injury in the hypertensive rat on 14D HS (8% NaCl). In contrast, the group supplemented with lysine had a significant reduction in KIM-1 staining. The scale bar is 100 μm; ($n = 6$ animals per group), error bar = SD. **E** Lysine proximal tubule metabolism is altered when kidneys are damaged. Different metabolite classes are color-coded. Metabolites changed with $q = 0.05$ in at least one experiment are depicted. In total, 89 metabolites were significantly altered. **F** Decrease of free fatty acids in hypertensive kidneys of lysine-treated animals. The boxplot shows a summary of the regulation of 22 quantified free fatty acids (both saturated and unsaturated). **G** Decrease of sugar metabolites glucose and di-hexose, and increase of NAD in lysine treatment in hypertension. Source data for this figure is available.

to a very high abundance of lysine to test if accelerated lysine metabolism was relevant in humans. The hypothesis was that healthy volunteers ("healthy") and hypertensive patients "at risk" for hypertension and CKD should respond differently in terms of metabolism. We selected "at risk" hypertensive patients that had likely kidney damage as a driving factor of hypertension. Four patients had unilateral kidney; thus, they are likely to develop salt-sensitive hypertension and proteinuria due to glomerular scarring and hyperfiltration[29–31]. An additional hypertensive patient had a mild proteinuria. Urine was collected for 12 h to see if key physiological data from discovery approaches in rats could be recapitulated in humans (Fig. 8A). In every healthy and at risk patient, an increase in albuminuria was observed (Fig. 8B), consistent with the blockage of albumin uptake and tubular proteinuria (Fig. 4). In the risk patients but not the healthy volunteers, urinary metabolome changes were observed. Consistent with the results from the animal model (Fig. 6), an increased abundance of modified lysine molecules (Ne-acetyllysine; saccharopine) was excreted with lysine treatment in risk patients (Fig. 8C). We also observed a reduction of excretion of Fumarate and Cis-aconitate, suggesting depletion of the TCA cycle, again only in the risk patients (Fig. 8C), which is consistent with the animal data (Figs. 6 and 7). No significant change in urinary arginine or other positively charged amino acids was observed (Fig. S8).

## Discussion
Metabolites are commonly viewed as biomarkers but are essential drivers of physiological adaptations. Alterations in lysine metabolism are widely observed in patients obtaining diets protective from hypertension[18–20], and lysine transport is also associated with kidney function across multiple GWAS and mGWAS studies. A high abundance of lysine in the urine (associated with variants in the tubular lysine uptake transporter SLC7A9) is associated with a good prognosis[17], and these findings were replicated in another CKD cohort[14]. The important role of lysine for kidney disease is also supported by animal studies suggesting that lysine supplementation does not alter lysine abundance in the serum but has effects on secondary disease entities in the bone[32]. Here, we explored the possibility that differential lysine metabolism can be a potential lever for a complex disease consisting of hypertension and kidney disease.

First, we developed a technology to utilize untargeted metabolomics workflows in order to quantify lysine fate across tissues. Isotope tracing methods usually use targeted metabolomics in order to develop hypothesis-driven views or models of metabolism. To catalog as many molecules as possible, we performed global untargeted metabolomics analyses to identify labeled metabolites[33,34]. This workflow compares lysine metabolites in an extension of previous studies that focused on protein incorporation[35–37]. The approach used here comes with some

limitations of untargeted metabolomics but also has several key advantages. First, one can catalog the entirety of metabolites of a distinct molecule at a broader depth as targeted approaches. Second, it allows the discovery of new and biologically important molecules from untargeted isotope-labeled metabolomics data. Labeling at very high percentages (over 95% of lysine molecules were labeled after 2 months of isotope diet) enables the identification of significantly incorporated metabolites in vivo. The feeding through the diet (as opposed to parenteral administration) allows potential analysis of microbiome-derived metabolites. The combination of untargeted isotope-based metabolomics and proteomics suggested that lysine is replaced faster with $^{13}C_6$ units before it is incorporated into the proteins and carnitines. This suggests a major network of lysine modification ("epimetabolome") independent of the incorporation of metabolites into proteins and that the main pool of free modified lysines derives from direct chemical and enzymatic reactions[5], and not from protein modifications such as histones.

We studied the Dahl salt-sensitive model since it showed a reduction of lysine metabolites and degradation products at the early onset of kidney damage and hypertension[22]. Interestingly, isotope tracing analysis revealed that Lysine metabolism was accelerated further in the salt-sensitive hypertensive rat model (Fig. 2). We found that this pathway could be exploited by using lysine as a metabolic diuretic. Lysine strongly decreased blood pressure, proteinuria, tissue damage, and increased $NAD^+$, a kidney-protective metabolite[38–40], while at the same time increasing diuresis. A primary mechanism is likely an increase in renal blood flow and glomerular filtration[41]. Mechanistically, these processes lead to diuresis that likely flushes out the protein plaques formed in the tubule system. Metabolically, a key finding was that administration of lysine leads to conjugation of lysine to central carbon molecules, while it prevents these molecules from modifying proteins. Several previous studies suggested the presence of detrimental lysine modifications in hypertension[42,43], from which lysine may protect. The excreted forms of lysine is likely also metabolically beneficial: one prominent and fast lysine overflow system is the lysine-conjugation to central carbon metabolites. Acetyl-CoA, and glucose, form acetyl-lysine and fructoselysine (Amadori reaction), respectively. The formation of saccharopine consumes NADH and alpha-ketoglutarate and generates NAD and Glutamate. Together, these processes are relevant for the depletion of the central carbon metabolism of the kidney. The observation of molecular Nε−malonyl-lysine, a previously unidentified metabolite, was only possible through the in vivo isotope labeling approach. It forms faster in hypertension and might be able to reduce ACC activity in vitro. It conveys—directly or indirectly—a protective effect on FFA metabolism. This alteration of FFA synthesis substrates by Nε-malonyl-lysine might further amplify the inhibition of albumin-bound FFA

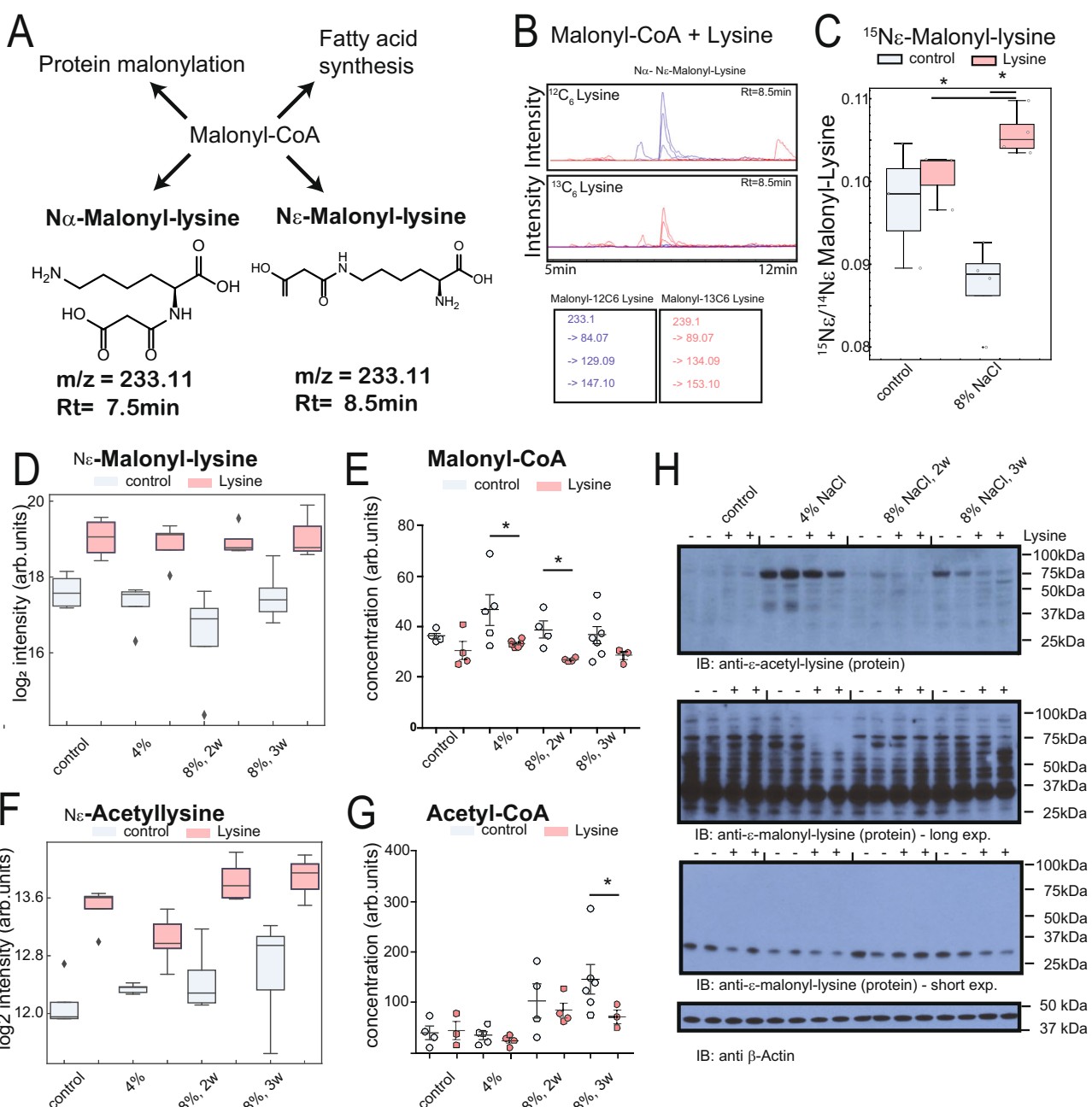

**Fig. 5 Previously undescribed metabolite malonyl-lysine connects lysine metabolism to reduced fatty acid synthesis. A** Structure and potential formation of Malonyl-lysine, a previously undescribed metabolite and potentially relationship to fatty acid synthesis. **B** Non-enzymatic formation of Nε-Malonyl-lysine by incubation of lysine and Malonyl-CoA. Both $^{13}C_6$ and $^{12}C_6$ lysine were used in the reaction, and detected by targeted metabolomics assay with heavy and light transitions. **C** Accelerated formation of Nε-malonyl-lysine in hypertension. Animals from Fig. 3D were analyzed for the formation of Nε-malonyl-lysine ($n = 3$ independent animals). **D** Nε-malonyl-lysine abundance in kidney cortices of hypertensive animals with and without lysine treatment ($n = 4$ independent animals for control, 8% 2w, and 8% 3w diet, and $n = 5$ for 4% diet, $p < 0.05$ in a two-tailed $t$-test). **E** Nε-malonyl-CoA abundance in kidney cortices of hypertensive animals with and without lysine treatment. Each dot is an observation from an independent animal, error bar = SEM. **F** Nε-acetyl-lysine abundance in kidney cortices ($n = 5$ independent animals, $p < 0.05$ in a two-tailed $t$-test). **G** Acetyl-CoA abundance in kidney cortices. Each dot is an observation from an independent animal, error bar = SEM. **H** Immunoblot analysis of protein acetylation and malonylation in cortex lysates as detected with respective antibodies. β-Actin serves as loading control. Representative run of in total $n = 2$ runs for $n = 4$ independent animals per group. Source data for this figure is available.

uptake, a general mechanism of fibrosis development[44]. Together, several central carbon metabolites can be depleted by lysine administration in the salt-sensitive background.

Translationally, the depletion of central carbon metabolites from the kidney is reminiscent of the effect of SGLT2 inhibitors. SGLT2i inhibitors lead to glucose excretion and are a recent breakthrough intervention to delay the outcome of cardiovascular kidney disease[45,46]. Lysine here is used as a vehicle to excrete other central energy equivalents. Notably, we observed that patients with vulnerable kidneys and risk for developing salt-sensitive hypertension react similarly to the rat model: they rapidly conjugate lysine to central energy metabolites, and excrete

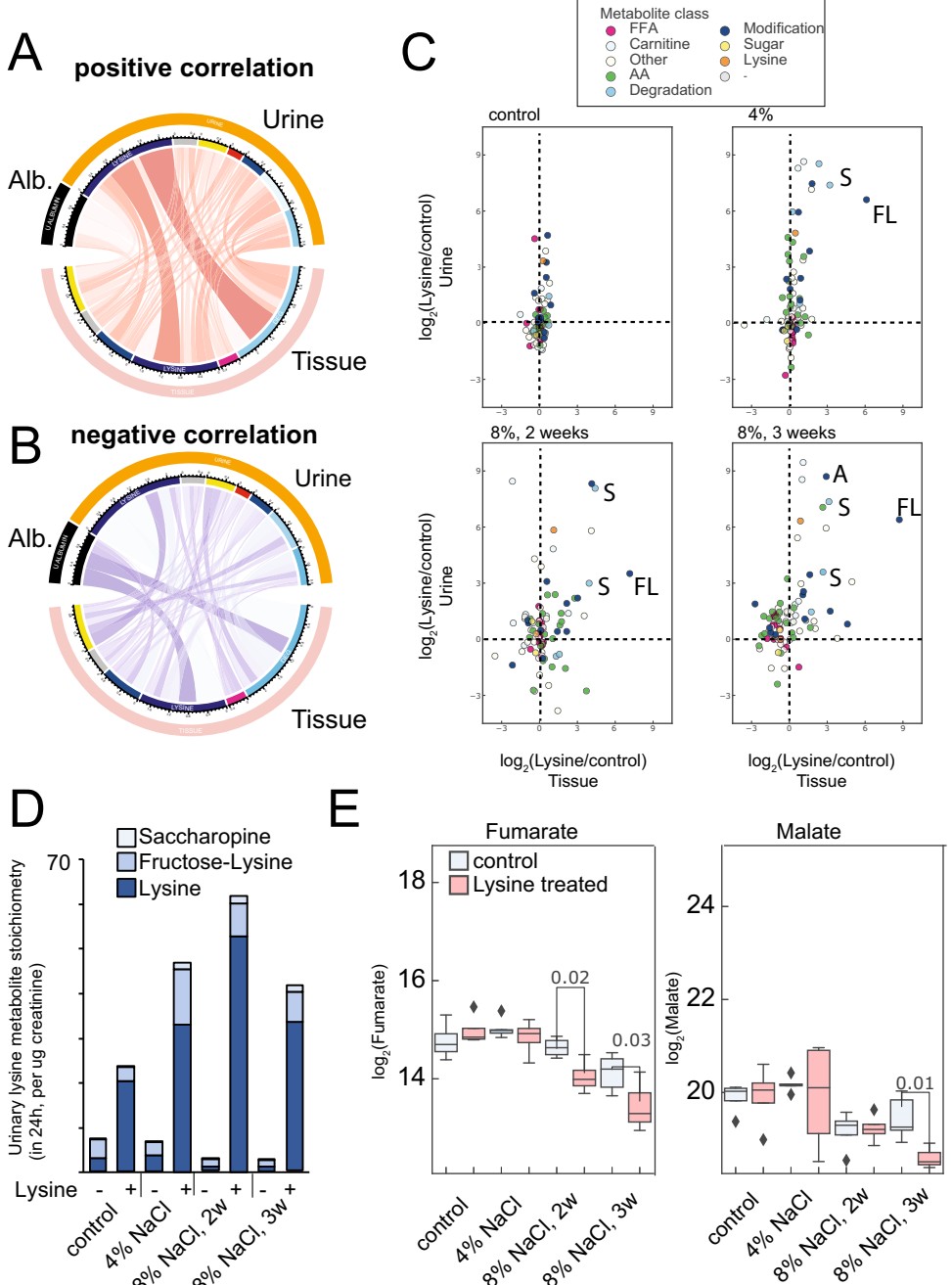

**Fig. 6 Urine is a sink for lysine conjugate metabolites. A** Integrative analysis depicting kidney and urine metabolite correlation. Chord diagram depicting positive correlations between kidney tissue metabolites and the respective urinary classes. **B** Chord diagram depicting negative correlations between kidney tissue metabolites and the respective urinary classes. The strongest negative correlations were (1) Albumin and Lysine in the cortex; (2) Albumin and lysine degradation products in the cortex, and (3) lysine and sugar metabolites in the cortex. **C** Scatterplot depicting $\log_2$ fold-changes (lysine vs control) in urinary metabolites and kidney cortex metabolites in untargeted metabolomics analysis. The different metabolite classes are color coded. **D** Urinary lysine metabolite stoichiometry as determined by targeted metabolomics analysis in 24 h collected urine normalized by urinary creatinine. **E** Depletion of terminal TCA cycle metabolites by lysine treatment as determined by targeted metabolomics analysis ($n \geq 4$ independent animals per group, two-tailed $t$-test). Source data for this figure is available.

it. Side findings are an overall decrease in the abundance of some central carbon metabolites such as Cis-aconitate, a proinflammatory metabolite. It should be noted that the proteinuria observed in these patients is likely a tubular proteinuria and not detrimental. It is likely caused by excessive lysine uptake in the proximal tubule. Patients with genetic defects in the lysine-inhibited albumin uptake mechanism have normal kidney function[47].

This study has limitations. The final kidney physiological mechanism and the exact bioactive molecule mediating lysine's protective effect, remain elusive. However, it is conceivable that not a single molecule might convey the effect, but the additive effects of metabolism and additional metabolite-proteome interactions. In addition, we cannot fully exclude further protective mechanisms on the level of electrolyte balance and kaliuria[48] or competition with other positively charged amino acids that may

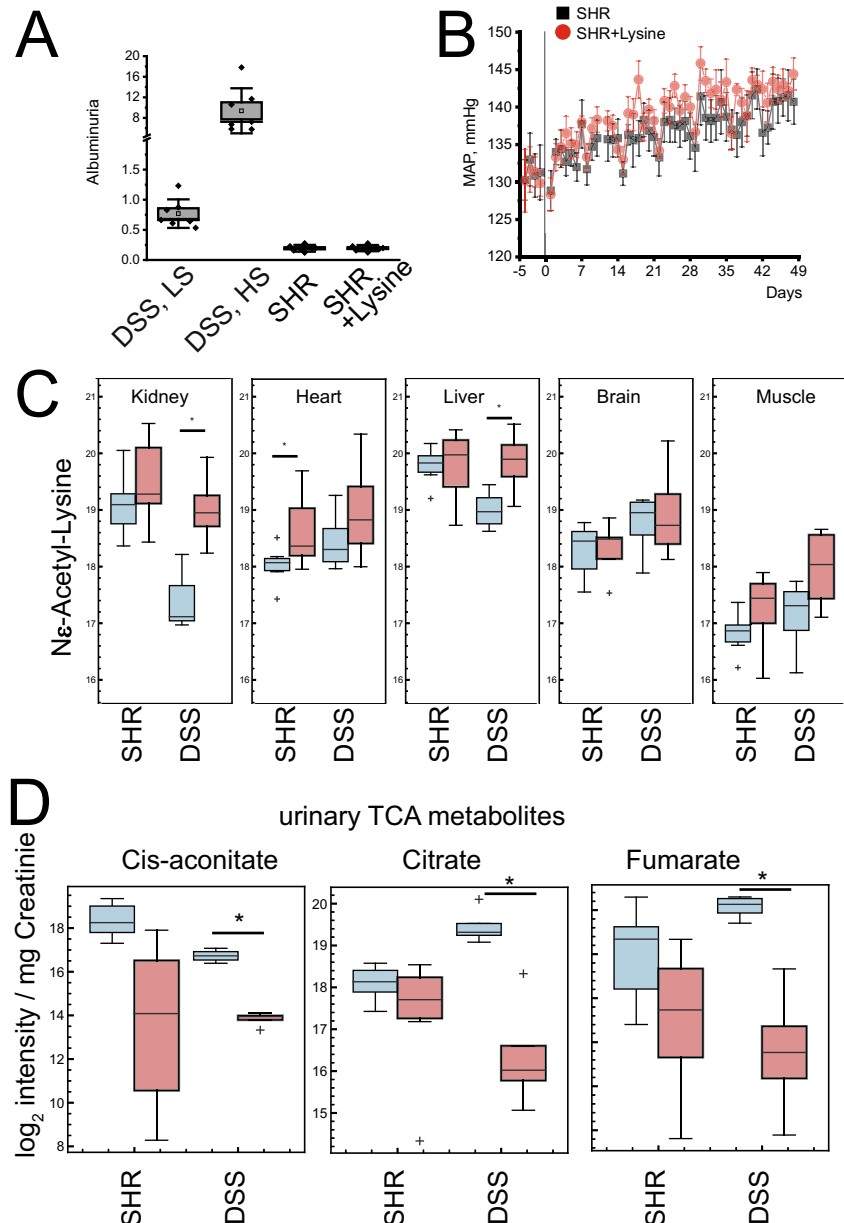

**Fig. 7 Lysine challenge does not develop protection or metabolic perturbation in spontaneous hypertensive rats (SHR), a hypertensive model without kidney disease. A** Proteinuria in SHR rats compared to DSS rats ($n = 6$ independent animals), error bars = SD. **B** Effect of Lysine supplementation on the mean arterial pressure (MAP) in SHR rats ($n = 6$ independent animals), error bars = SD. **C** Effect of lysine supplementation on the formation of Nε-acetyllysine. Kidney and liver Nε-acetyllysine increases in D/SS rats with comparable hypertension, but not in SHR rats ($n \geq 4$ animals per group). **D** Effect of lysine supplementation on urinary TCA cycle metabolome ($n \geq 4$ animals). Source data for this figure is available.

have bioactivity, such as arginine, although our cursory studies did not show effects on these parameters. Finally, the observed pathways may be species-dependent and might not necessarily reflect metabolic physiology in humans. A pilot study in humans showed comparable effects, but, for instance, the identity of significantly changed TCA metabolites in urine was only partly overlapping in rats and humans at risk. However, the metabolomics perturbations integrate with organ physiology, increasing the need to understand further protein-metabolite interplay in complex diseases such as kidney disease.

In conclusion, the analysis of labeled metabolites demonstrated the kidney-specific role of lysine metabolites and their physiologic activity. Kidney protective effects in hypertension occur via lysine's unique physiology in the tubules and its chemically modified entities that provide a sink for important

sugar metabolites of the central carbon and fatty acid metabolism. Thus, lysine's metabolic activity alters the epimetabolome for kidney protection (Fig. 8D).

## Methods

**$^{13}C_6$ Lysine isotope labeling study.** Male Bl6/N mice (12-weeks old) were fed a custom diet (Silantes) containing more than 99% $^{13}C_6$ lysine. The protocol for labeling was described previously[35]. After 1, 2, and 8 weeks (protocol 1) or 1, 2, and 3 weeks (protocol 2), mice were sacrificed and perfused with ice-cold PBS. All denominated organs were snap-frozen from the same mouse and stored at −80 °C degrees. Blood was separated into erythrocyte and plasma by centrifugation.

**Metabolomics sample preparation.** Frozen organs were kept on dry ice and 10 mg of tissue was weighed. Then, 800 μl ice-cold extraction solution containing acetonitril:methanol:water 2:2:1 was added. Samples were subjected to tissue homogenization using a multiplex-bead-beater (Storm) for 30 s (liver), 1 min

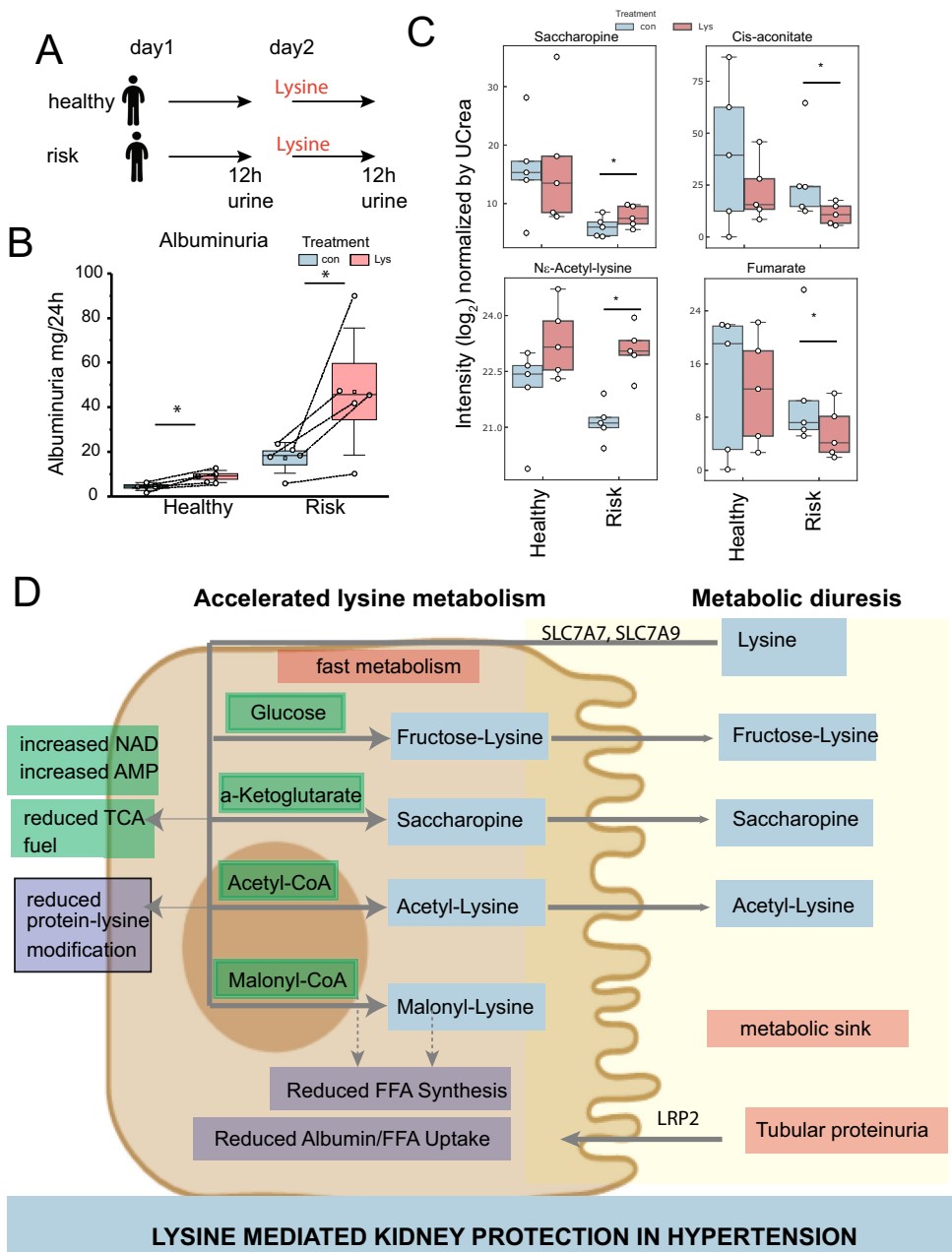

**Fig. 8 Lysine's metabolic physiology is altered in humans at risk for hypertensive kidney disease. A** Pilot study design. Kidney healthy volunteers and patients at risk for development of hypertensive kidney disease (solitary kidney, or hypertension and mild proteinuria of <1 g/24 h) were included and subjected to oral lysine, followed by 12 h urine collection. **B** Albuminuria in healthy and risk patients ($n = 5$ independent patients per group) induced by lysine. * <0.05 in a paired $t$-test. **C** Metabolomic analysis of urine of healthy and risk patients. Reduction of TCA cycle metabolites was observed while modified lysine metabolites were increased. Intensities are normalized to UCreatinine, $n = 5$ independent patients per group, two-tailed paired $t$-test. **D** Summary depicting discovery of lysine-dependent kidney protection by proximal tubule protection, metabolome alterations due to shifting cellular metabolism and sinking central carbon conjugate metabolites in the urine. Source data for this figure is available.

(lung), or 45 s (all other tissues). The supernatant was transferred to a prechilled Eppendorf tube. Then, beads were washed with 200 μl of the extraction solution, and the wash solution was added to the remaining homogenate. The homogenate was incubated at −20 °C for 2 h. Then, tissue was spun down at 4 °C for 20 min, and the supernatant was transferred to another vial. The supernatant containing the extract was transferred into a speed vac and dried down. The next morning, the dried-down extract was once resuspended in 100 μl (per 10 mg tissue) of acetonitrile-water 1:1, and the solution was centrifuged at 4 °C. Then, the solution was transferred to autosampler vials and stored at −80 °C until further use.

**Untargeted metabolomics analysis and mass spectrometry.** LC-MS/MS analysis for metabolomics was performed as previously described[22]. Data were

annotated with in-source fragments and adducts. Quality controls were run every five samples. The mass spectrometer was initially calibrated using NaFormate peaks and in addition post-run. For untargeted metabolomic analysis, we used a UHPLC-MS approach. For fractionation, we used hydrophilic interaction liquid chromatography (HILIC) fractionation and reversed-phase (RP) chromatography as previously described[22]. We used a quadrupole time-of-flight instrument (Impact II, Bruker, Bremen, Germany) coupled with an ultrahigh-performance liquid chromatography (UHPLC) device (Bruker Elute, Bruker, Billerica, MA), or to an Agilent Infinity 1290 UHPLC device (Agilent, USA). The MS was calibrated using sodium formate (post-run mass calibration). Data were acquired over an $m/z$ range of 50–1000 Da in positive ion mode and negative ion mode (HILIC only). Electrospray source conditions were set as

follows: end plate offset, 500 V; dry gas temperature, 200 °C; drying gas, 6 L/min; nebulizer, 1.6 bar; and capillary voltage, 3500 V.

To increase metabolome coverage and minimize ion suppression, we used a dual fractionation strategy. For RP separation, an ACQUITY BEH C18 column (1.0 × 100 mm, 1.7-µm particle size; Waters Corporation, Milford, MA) was used, and for HILIC fractionation, a ACQUITY BEH amide (1.0 × 100 mm, 1.7-µm particle size; Waters Corporation, Milford, MA) column was used. The flow was 150 µl/min, and a binary buffer system consisting of buffer A (0.1% FA) and buffer B (0.1% FA in acetonitrile) was used. The gradient for RP was: 99% A for 1 min, 1% A over 9 min, 35% A over 13 min, 60% A over 3 min, and held at 60% A for an additional 1 min. The gradient for HILIC consisted of 1% A for 1 min, 35% A over 13 min, 60% A over 3 min, and held at 60% A for an additional 1 min. The injection volume was always 2 µl. For molecule identification purposes, putative molecules of interest were fragmented using three different collision energies (10, 20 eV) or ramp collision energies (20 to 50 eV).

**Untargeted metabolomics data analysis**. Bruker Raw files (*.d) were transformed into mzml files using the compassxport_converter.py script (Bruker). Then, data files were uploaded to XCMS online, and differential peaks were extracted in positive and negative ion mode[49]. Feature detection was performed with the centwave method with the following options: ppm = 10, minimum peak width = 5, maximum peak width = 20, mzdiff = 0.01, signal/noise = 6, integration method = 1, prefilter peaks = 3, prefilter intensity = 100, noise filter = 100. For retention time correction, we used obiwarp method with profStep = 1. Alignment was performed with bw 05, minfrac = 0.5, mzwid = 0.015, mminsamp = 1, and max = 100. Statistical test was performed using an unpaired parametric t-test (Welsh), with post-hoc analysis = TRUE,. Statistical filtering was performed for prioritization of features of interest, including a fold change of at least 1.5, and a p-value < 0.05, and a corrected p value (q-value) <0.05. The analysis included PCA and multivariate analyses. Metabolites were identified based on (1) unique mass, (2) MS[2] spectra comparison to authentic standard (3) coelution with authentic standard, and (4) isotopic pattern. The following adducts were routinely considered: Na, H, for positive mode, and Cl, formate for negative mode. Intensity data were plotted using instantclue[37], ggplot2, or circosplot.R package.

**Kidney dissection**. Kidney cortices were isolated under manual control (protocol 1, Fig. 1). Kidneys were manually dissected using a stereomicroscope. Based on anatomical criterions, the kidneys were dissected into cortex, inner stripe outer medulla, outer stripe outer medulla and inner medulla (protocol 2, Suppl. Fig. 3). The samples were snap-frozen and stored at −80 °C or dry ice.

**Proteomics sample preparation**. Kidney samples of $^{13}C_6$ labeled kidneys were subjected to proteomics analysis using a tryptic in solution digestion protocol followed by nLC-MS/MS analysis. In brief, kidney pieces were minced and homogenized using a glass homogenizer in 8 M urea containing 10 mM ammoniumbicarbonate as well as protease and phosphatase inhibitor cocktail (Thermo). The homogenate was spun down at 6 °C for 20 min, and the supernatant was kept for further analysis. A small aliquot was subjected to protein measurement using BCA assay (Thermo). The proteins were reduced and alkylated using 5 mM DTT (30 min) and 10 mM IAA (1 hr) in the dark. Then, urea concentration was diluted to <2 M and LysC (1:100 w/w) ratio was added, and the mixture was digested for 16 h at 37 °C. The next day, the reaction was terminated by the addition of 2% formic acid. Peptides were desalted using in-house made stage-tips and analyzed by nLC-MS/MS.

**Proteomics analysis of $^{13}C_6$-lysine treated mice**. The peptides were separated by reverse-phase nanoflow-LC-MS/MS analysis and sprayed into a quadrupole-orbitrap tandem mass spectrometer (qExactive plus, thermo scientific) as previously described[50]. Raw Proteomics data were parsed with MaxQuant v 1.5.3.3.[51], using a Uniprot RefSeq reference proteome database (www.uniprot.org) from Jan 2017, with using LysC (cuts after each Lysine) as a protease. Multiplexicity of the analysis was 2, with $^{13}C_6$ labels in proteins as a modification in the second channel. The analysis has been previously described[52]. The non-normalized ratios (Heavy/$^{13}C_6$ over Light$^{12}C_6$) were used for further analysis using Perseus[53] software suite and filtering for ratios in all experiments, as well as for annotation with GO terms.

**Dahl salt-sensitive (D/SS) rats**. The strain of re-derived Rapp Dahl SS rats used in studies (SS/JrHsdMcwi, RRID:RGD_1579902) has been inbred for more than 50 generations at the Medical College of Wisconsin. Male and female animals at the age of 8 weeks were used for experiments. Rats were maintained on AIN-76A custom diet, either low salt (LS; 0.4% NaCl, #113755, Dyets Inc.) or high salt (HS; 4% or 8% NaCl, #113756 or #100078, respectively, Dyets Inc.). Water and food were provided ad libitum.

D/SS rat is a widely used model of salt-induced hypertension and CKD. Since the derivation of the D/SS rat in 1962, there have been numerous phenotyping studies demonstrating the importance of the kidney in the regulation of blood pressure. Cowley et al. showed that upon consuming a high salt diet, the D/SS rat rapidly becomes hypertensive and exhibits severe renal damage, yet GFR is not

changed until 10 days after consuming a high salt diet[54]. Housing conditions were 12 h light-dark-cycle in a temperature and humidity controlled facility.

**Lysine treatment study**. Experimental animals received either vehicle (water) or L-Lysine (17 mg/ml) via drinking water (n = 6 per group). Male or female D/SS rats were anesthetized with 2–3% (vol/vol) isoflurane and a blood pressure transmitter (PA-C40; DSI) was surgically implanted subcutaneously, with the catheter tip secured in the abdominal aorta via the femoral artery. After a 3-day recovery period, blood pressure was measured with a DSI system ("telemetry") in conscious, freely moving SS rats under HS diet protocol, similar to those described previously[55–57].

For urine collection, rats were placed in metabolic cages (no. 40615, Laboratory Products) for a 24 h urine collection. These urine samples were used to determine electrolytes, microalbumin, and creatinine. Whole blood and urine electrolytes and creatinine were measured with a blood gas and electrolyte analyzer (ABL system 800 Flex, Radiometer, Copenhagen, Denmark)[55]. Kidney function was determined by measuring albuminuria using a fluorescent assay (Albumin Blue 580 dye, Molecular Probes, Eugene, OR) read by a fluorescent plate reader (FL600, Bio-Tek, Winooski, VT).

**Pulsed $^{15}N$ lysine labeling in the D/SS rat**. Male D/SS rats on HS and LS protocol were administrated with $^{15}N$ L-lysine-2HCl (Cambridge Isotope) at day 13 HS diet (8% NaCl; 24 h before sacrifice). Intraperitoneal injection of $^{15}N$ L-Lysine-2HCl (340 mM; 200 µl of PBS solution) was performed 24 h before sacrifice. Urine was collected for 24 h after injection.

**Albumin uptake cell culture studies**. Confocal microscopy was used to detect uptake of fluorescent albumin (AlexaFluor-647 albumin, 40 µg/ml) in confluent monolayers of OK proximal tubule epithelial cells[58] (visualized by fluorescent F-actin, 488 nm), grown on transwell filters under orbital shear stress as previously described and coincubated with lysine or glycine[59]. Cells were incubated for 1 h in serum-free culture media containing fluorescent albumin), after pretreatment of 5–50 mM L-Lysine (or other amino acids) for 2 h, or after overnight treatment with 1–10 mM L-lysine. Treatment was performed from the apical side unless otherwise indicated. Cell-associated albumin was quantified by spectrofluorimetry.

**Intravital dual photon microscopy**. All surgical and imaging procedures were performed as described previously[60,61]. Briefly, imaging was conducted using an Olympus FV1000 microscope adapted for an intravital two-photon microscopy with high-sensitivity gallium arsenide nondescanned 12-bit detectors. Animals were anesthetized with pentobarbital sodium (50 mg/ml). A jugular venous line was used to introduce fluorescent rat albumin (Texas Red labeled) and high-molecular weight dextran (150 kDa FITC-labeled, TdB Consultancy, Uppsala, Sweden).

**Immunohistochemistry**. Rat kidneys were fixed in 10% formalin and processed for paraffin embedding as previously described[62]. Kidney sections were cut at 4 µm, dried, and deparaffinized for subsequent labeling by streptavidin-biotin immuno-histochemistry. After deparaffinization, slides were treated with a citrate buffer (pH 6) for total of 35 min. Slides were blocked with a perioxidase block (Dako, Coppenhagen, Denmark), avidin block (Vector Laboratories, Burlingame, CA), biotin block (Vector Laboratories), and serum-free protein block (Dako). Tissue sections were incubated for 90 min in antibody to Kidney Injury Molecule-1 (Rat KIM-1Ab, 1:300, #AF3689, R&D Systems, Inc) or megalin (lipoprotein-related protein 2 (LRP2) Ab, 1:2500, from Dr. Franziska Theilig University of Kiel, Kiel, Germany). Secondary detection was performed with goat anti-goat or anti-rabbit biotinylated IgG (Biocare, Tempe, AZ) followed by streptavidin-horseradish peroxidase (Biocare) and visualized with diaminobenzidine (Dako). All slides were counterstained with Mayer's hematoxylin (Dako), dehydrated, and mounted with a permanent mounting medium (Sakura, Torrance, CA).

**Bioinformatic image analysis via convolutional neural net analysis**. Rat kidneys were cleared of blood, formalin fixed, paraffin embedded, sectioned, and mounted on slides as previously described[63]. Slides were stained with Masson's trichrome stain. The localization and scoring of glomeruli was performed by a robust application of convolutional neural nets[23,64]. A cumulative distribution plot was generated (OriginPro 9.0) using glomerular injury scores based on a scale of 0–4 as previously described[65], and the probability for a corresponding score interval was calculated (more than 3500 glomeruli per group). Cortex protein cast analysis was performed using a color deconvolution filter and Analyze Particles mode in the Fiji image application (ImageJ 1.51 u, NIH).

**Bioinformatic algorithms for mass-difference-based isotope selection**. Labeled isotopologues were detected using an in-house isotracker script (see Data availability and github submission) that operated as follows: first, we search for groups of features that, according to their m/z difference (<10 ppm), could correspond to an isotopic envelope composed of a light isotope and heavy isotopes. Only features within 2 s of retention time difference were allowed to be grouped into a single

isotopic envelope. Next, we compared the isotopic envelope between labeled and unlabeled samples. The envelopes presenting at least one heavy isotope with statistically significant higher abundance in labeled samples compared to unlabeled samples were retained for further analysis.

**Correlation-based isotope selection approach.** Labeled isotopologues were detected using an in-house script (see Data Availability and github submission) that operated as follows: The custom R script is designed to identify isotopes from stable labeled isotope LC-MS data based on an input list of compounds and sum formulas and is available online including documentation via https://github.com/hpbenton/targeted_isotopes. It uses mzR and MSnBase from the bioconductor repository to open and manipulate the data. Once the raw data is opened each file is independently searched for a list of possible compound hits. These compounds are searched by creating a small bin of a user chosen ppm range around the mass. The vector of data undergoes a smoothing using a Savitzky Golay filter. Any compound that is above a given threshold (default 1000 counts) and is also above the chosen signal to noise is selected. Then, since the formula is known, any and all isotopes are searched within the same ppm range and at that retention time range. If the isotope peak also satisfies the above criteria the two vectors are correlated to help confirm a true positive isotope. Most isotopes are correlated above 0.9[66], Supplementary Fig. 1. The script used a list of 400 lysine metabolites derived from KEGG and METLIN as input.

**Synthesis of Nε-malonyl-lysine.** For a detailed description of Nε-malonyl-lysine and Nα-malonyl-lysine isomers, please see the supplementary material and methods. Molecules were synthesized as described in the supplementary material and methods and characterized by NMR and mass spectrometry. Volatiles were removed under reduced pressure and the crude product was purified by mass-directed preparative reversed-phase HPLC to give the formic acid salt which was directly used as an analytical standard and fragmented at 10, 20, and 20–50 eV in ESI in positive ion mode.

**Malonyl-CoA and lysine in vitro reaction.** 1 μM lysine and a 10 μM of malonyl-CoA was incubated together in 10 μl of PBS, pH = 8 at 37 degrees for 1 h. Both the isotope-labeled ($^{13}C_6$) and non-isotope-labeled ($^{12}C_6$) form of lysine were used, in order to exclude unspecific molecule products. Mixtures were analyzed on a QQQ instrument. Specific transitions were used in order to detect both heavy and light forms of N-e-malonyl-lysine as well as lysine, and malonyl-CoA.

**Malonyl-CoA assay.** Malonyl-CoA levels from tissue lysates were determined using a commercial rat ELISA assay (MyBiosource.com) according to the manufacturer's manual. ACC assay was performed from lysates of OK proximal tubule cells according to the manufacturer-s instructions.

**Targeted metabolomics.** Targeted metabolomic analysis was performed on a triple-quadrupole (QQQ) mass spectrometer (Agilent Triple Quadrupole 6490, San Diego, CA), and the LC part was coupled to a high-performance liquid chromatography (HPLC) system (1290 Infinity, Agilent Technologies) coupled to ion funnel. For glycolysis and TCA product metabolite, a ZIC-pHILIC (Sequant column; $2.1 \times 150$ mm) was used for separation. Cycle time was 100 ms. Collision energies and product ions (MS2 or quantifier and qualifier ion transitions) were optimized. Electrospray ionization source conditions were set as follows: gas temperature, 250 °C; gas flow, 12 L/min; Nebulizer, 20 psi; sheath gas temperature, 350 °C; cap voltage, 2000 V; and nozzle voltage, 1000 V. The gradient consisted of buffer A and buffer B. Buffer A was 95:5 $H_2O$:acetonitrile, 20 mM $NH_4OAc$, 20 mM $NH_4OH$ (pH 9.4). Buffer B was acetonitrile. The gradient with A/B ratios were as follows: T0, 10:90; T1.5, 10:90; T20, 60:40; T25, off. Five microliters were injected. For analysis of lysine metabolites, identical column and chromatography conditions as in the "untargeted metabolomics" section were used. In all cases, a standard curve was recorded and integrated using the mass hunter platform (Agilent). The method for the TCA cycle including transitions in tissue was previously published[22]. The transitions used for malonyl-Lysine were as follows: 233 -> 84; 233 -> 129.09, 233 -> 147.10 (for non-labeled malonyl-Lysine) and 239 -> 89, 239 -> 134, 239 -> 153 for 13C6 labeled malonyl-Lysine).

For the TCA cycle analyte measurements in urine, we used a separation on a CSH-Phenyl-hexyl column, 100 mm length and 1 mm diameter (Waters) at flow rates of 150 μL/min as previously described[67]. Here, targeted metabolomic analysis was performed on a Triple-Quadrupole (QQQ) mass spectrometer (Agilent Triple quadrupole 6490, San Diego, CA) coupled to a HPLC system (1290 Infinity, Agilent Technologies) coupled to ion-funnel. A Phenyl/Hexyl CSH column (1.7 μm, $1.0 \times 100$ mm) (Waters, Taastrup, Denmark) was used for separation. Dynamic multi-reaction monitoring (dMRM) was used in which the cycle time was set by the system. The collision energies and product ions (MS2 or quantifier and qualifier ion transitions) were optimized. ESI source conditions were set as follows: Gas temperature 290 °C, gas flow = 13 L/min, Neb = 35 psi, sheath gas temp 350 °C, sheath gas flow 12 L/min, cap voltage 2000 V, and nozzle voltage 1500 V. The gradient was consisting of buffer A and buffer B. Buffer A was 99.9% $H_2O$ and 0.1% formic acid. Buffer B was 99.9% acetonitrile and 0.1% formic acid. The Gradient with A/B ratios were as follows: T0:99/1, T0.5: 99/1, T3.5: 75/25; T3.6:

0:100, T5.0 0:100, T5.1: 99/1 off. 2 μl were injected. The used transitions for metabolites can be found in Supplementary Data 5.

**Immunoblot.** Protein samples in RIPA buffer were loaded onto Novex 4–12% Bis-Tris gels (Life Tech) and were transferred onto nitrocellulose membranes with the Novex semi-dry transfer apparatus (Life Tech). After blocking in 5% milk-TBST for 1 h at room temperature, blots were incubated overnight in 5% BSA-TBST (1:1000 acetylated-lysine CST, 1:1000 malonyl-lysine CST, 1:5000 beta-actin Genscript A00702) at 4 deg Celsius. After washing in TBST, blots were incubated in 1:5000 HRP-conjugated secondary antibodies (mouse anti-rabbit Jackson Immunoresearch 211-032-171, rabbit anti-mouse Jackson Immunoresearch 211-035-109) in 1% milk-TBST. Blots were incubated with ECL Western blotting substrate (Pierce Scientific 32106) and were processed by autoradiography. The used ladder was Biorad Precision Plus All-blue ladder.

**Proteomics sample preparation.** Kidney samples were homogenized using stainless steel beads in a Bullet Blender STORM Pro device (Next Advance) with 6 M guanidine hydrochloride, 0.1 M HEPES pH 7.4, 5 mM EDTA buffer, complemented with protease inhitor cocktail (Roche). Protein homogenates were treated with Benzonase (Millipore) at 37 °C (15 min) to degrade nucleic acids and subsequently carbamidomethylated with 10 mM TCEP and 50 mM CAA at 95 °C (5 min). The concentration of proteins was determined using the BCA assay (Thermo) and 50 μg aliquots were purified with paramagnetic, mixed 1:1 hydrophobic:hydrophilic SP3 beads. Purified proteins were resuspended in 50 mM HEPES, pH 7.4 and digested over night at 37 °C with trypsin (Serva) in a 1:100 (w/w) ratio. Samples were acidified with 2% formic acid and peptides were purified using in-house made stage-tips.

**MS measurement of proteomics samples.** Samples were separated on an Ultimate3000 RSLC nanoHPLC coupled on-line to an Exploris480 orbitrap tandem mass spectrometer (Thermo). The HPLC was operated in a two-column setup with an Acclaim 5 mm C18 cartridge pre-column (Thermo) and a self-packed C18 column emitter. Separation was performed at 400 nL/min in a heated column oven at 50 °C (Sonation) with the following gradient of solvents A ($H_2O$ + 0.1% FA) and B (ACN + 0.1% FA): 5 min from 2–8% B, 80 min from 8–25% B, 10 min from 25–35% B and a high-organic washout at 90% B for 8 min followed by a re-equilibration to the starting conditions (2% B). The mass spectrometer was operated with the FAIMS device at standard resolution with a total carrier glas flow of 3.8 L/min at two voltages: −45 and −65 V. The Orbitrap resolution for the MS1 full scan was set to 60k, where as the MS2 scans were recorded with 1 s cycle time for each FAIMS CV at an orbitrap esolution of 15k with an isolation window of 1.4 $m/z$. Dynamic exclusion mode was set to custom with a 40 s exclusion window and a mass tolerance of 10 ppm each. Raw FAIMS data was converted into MzXML files with the FAIMS_MzXML_Generator tool (v1.0.7639[68]) and queried with MaxQuant v 1.6.7.0 (FDR = 1%, match between runs = on) using the UniProt reference proteome database for rat from May 2020 (canonical only, 21587 entries) and default settings for orbitraps. Enzyme specificity was set to Trypsin/P, cysteine carbamidomethylation was set as a fixed modification (+57.021464) and methionine oxidation (+15.994914) as well as protein N-terminal acetylation (+42.010565) were set as variable modifications. Data analysis was performed using Perseus software suite (v. 1.4.1.3)[53].

**Isolated tubule uptake study.** Tubules from male and female WT Bl6N mice were isolated using a previously published isolation protocol[50], and incubated with 15Ne-lysine (obtained from Cambridge isotope) at a concentration of 1 mM for indicated time courses. Metabolites were analyzed using HILIC chromatography on a BEH-amid column as described above on a QQQ triple-quadrupole mass spectrometry.

**Human research.** Informed consents was obtained from all study participants. Human studies were done in accordance with and approval of the Internal Review Board at the Medical College of Wisconsin and in the agreement of DHHS rules and regulations for human research. The study design and the use of human study participants was conducted in accordance with the criteria set by the Declaration of Helsinki. The project was approved by the MCW/FH IRB (first approved on 7/10/2017), with the project ID PRO00029215. To test the physiologic response to oral lysine in humans, we conducted the following pilot study. We tested the protocol on two groups of patients: normal subjects and subjects with a single kidney or hypertension. The healthy volunteers had no medical diagnosis and were 25–45 years old. The risk patients included four patients with unilateral kidney due to unilateral renal agenesis (2) and nephrectomy for kidney transplantation (2), and one patient with mild albuminuria (<1 g/24 h). None of the risk patients was obese, and age range was between 51 and 57. Two 12-h urine collections were done under the same conditions during daytime to avoid circadian variation. Two hours before starting the second collection, subjects were given 30 g of lysine hydrochloride (Now-Foods) dissolved in 250–300 ml of water or lemonade. Subjects were instructed to drink the lysine over 2–3 hours. The 2 collected urine samples were analyzed for urine creatinine, urine albumin using Siemens DCA-Vantage analyzers according to manufacturer recommendation. When urinary albumin

concentration was below the device/reagent sensitivity, the samples were concentrated using Millipore microfiltration units with a molecular weight cutoff of 3 kDA.

**Study in SHR rats**. The spontaneously hypertensive (SHR) rats were purchased via Charles River labs. Animals were fed a regular salt diet (no. 5001, LabDiet, Purina) with water and food provided ad libitum. Experimental animals received either vehicle (water) or L-Lysine (17 mg/ml) via drinking water ($n = 6$ per group). Blood pressure was continuously measured by telemetry for 7 weeks. In addition, urine collections and albuminuria were performed as described for the D/SS rats. The SHR rat is a well-established model of hypertension that spontaneously develops blood pressure[27]. In addition, the SHR rats develop much less renal damage than the stroke-prone SHR (SHRsp) or D/SS strains after salt-supplementation[28].

**Measurements of RAAS hormones by Attoquant**. Ang I (1–10), Ang II (1–8), Ang 1–7 Ang 1–5, Ang III (2–8), and Ang IV (3–8) (F) were quantified from plasma samples collected from Dahl S HS-treated (black) and HS-treated supplemented with lysine (red) rats ($n \geq 8$ for each group). ACE plasma activity was calculated by the ratio of Ang II to Ang I. Plasma renin activity (PRA) was represented by the total detected amount of Ang I and Ang II. Bounds of boxes represent SEM; whiskers span the standard deviation; median is denoted by a horizontal line; and the median is denoted by a square within each box. The quantification of RAAS components was performed in equilibrated heparinized plasma samples by Attoquant Diagnostics (Vienna, Austria) as was previously described[69,70].

**Study approval**. All studies using D/SS and SHR rats were conducted at the Medical College of Wisconsin and protocols were approved by the MCW Animal Care and Use Committees and were performed in accordance with the standards set forth by the NIH Guide for the Care and Use of Laboratory Animals (National Academies Press, 2011). All mice studies were conducted at the MPI for Heart Lung and Blood Research Bad Nauheim as previously described in accordance with the local authorities and regulations (Regierungspraesidium Darmstadt, and local animal ethics committee at the MaxPlanck Institute Bad Nauheim)[36,71,72].

**Statistical analysis**. Two-tailed *t*-tests were used for comparison unless otherwise indicated. Analysis of metabolomics and proteomics data is described in the respective sections. Statistical representation of box plots are as follows: The line in the box is the median represents the 50% quantile. The upper and lower limit show the 25 and 75% quantiles. The whiskers display the max and min values (1.5 times the inter quantile range). All data analysis of phenotypic data with reported statistics and p-values was performed using two-tailed unpaired *t*-tests unless otherwise reported.

**Reporting summary**. Further information on experimental design is available in the Nature Research Reporting Summary linked to this article.

## Data availability

Source data is provided with the paper. Proteomics data is available through the PRIDE/proteomExchange repository[73,74], http://www.ebi.ac.uk/pride. Project accessions: PXD007749, PXD029232. Metabolomics data are available through Massive[75]. MSV000089224, MSV000089223. Uniprot RefSeq reference proteome databases from January 2017 (mouse) and from May 2020 (rat) were downloaded from www.uniprot.org. All the other data supporting the findings described in this manuscript are available in the article and in the Supplementary Information and from the corresponding author upon reasonable request.

## Code availability

Processing scripts are available through Github (see isotope selection approaches, for correlation-based approach https://github.com/hpbenton/targeted_isotopes), and for mass-difference approach https://github.com/xdomingoal/isoTracker), https://doi.org/10.5281/zenodo.6447658.

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

## Acknowledgements

We would like to thank Vladislav Levchenko and Denisha Spires for their help with collection of materials and work with animals, John Bukowy (all Medical College of Wisconsin) for help with the application of convolutional neural nets for the glomerular damage analyses, Rikke Nielsen (Aarhus University) for critically reading the manuscript, Ruben Sandoval and Bruce Molitoris (Indiana Center for Biological Microscopy) for help with two-photon imaging, Mogens Johannsen and Troels Rønn Kjær (Aarhus University) for help with enzyme analysis. The authors acknowledge the help of the CECAD proteomics core, and the technical assistance of Ruth Herzog and Mette Løbner. This research was supported by the National Institutes of Health grants HL135749 (to A.S), CA231991 (to B.F.C.), DK126720 (to O.P.), UL1TR001450/SCTR 2214 (to O.P.), endowed funds from the SC SmartState Centers of Excellence (to O.P.), Spanish Ministry of Science and Innovation — State Research Agency (AEI) grant PID2019-106277RA-I00 (to X.D.-A.), and Department of Veteran Affairs I01 BX004024 (to A.S.). M.M.R. was supported by the DFG (Ri2811-1/2, Ri2811-2), and the work is currently supported the Novo Nordisk Foundation Young Investigator Grant to M.M.R., as well as Aarhus University Forskening Fonden (AUFF). This project has received funding from the European Union's Horizon 2020 research and innovation programme under the Marie Skłodowska-Curie grant agreement No 754513 and The Aarhus University Research Foundation (AIAS-COFUNDII fellowship to M.M.R.).

## Author contributions

Performed experiments. M.M.R., O.P., L.D., D.G., A.P., M.L.G., M.H., N.H., B.P.K., F.D., J.J., and A.E.-M. Contributed new reagents/tools: M.A.S., N.H., M.B., B.F.C., M.K., H.P.B., and M.H. Analyzed data: M.M.R., O.P., D.G., X.D.-A., A.P., C.G., T.B., J.X., M.B., H.P.B., and A.E.-M. Interpreted data and discussion: M.M.R., O.P., A.E.-M., T.B., M.B., E.S., O.A.W., G.S., and A.S.

## Competing interests

The authors declare no competing interests.

## Additional information

[1]Scripps Center for Metabolomics, Scripps Research, La Jolla, CA 92037, USA. [2]Department of Biomedicine, Aarhus University, Aarhus, Denmark. [3]III. Medical Clinic, University Hospital Hamburg Eppendorf, Hamburg, Germany. [4]AIAS, Aarhus Institute of Advanced Studies (AIAS), Aarhus University, Aarhus, Denmark. [5]Division of Nephrology, Department of Medicine, Medical University of South Carolina, Charleston, SC 29425, USA. [6]Division of Nephrology, Department of Medicine, Medical College of Wisconsin, Milwaukee, WI 53226, USA. [7]Omics Sciences Unit, EURECAT, Technology Centre of Catalonia, Reus, Catalonia, Spain. [8]Department of Molecular Pharmacology and Physiology, University of South Florida, Tampa, FL 33602, USA. [9]Department of Physiology, Medical College of Wisconsin, Milwaukee, WI 53226, USA. [10]Department of Chemistry, Scripps Research, La Jolla, CA 92037, USA. [11]Renal Electrolyte Division, Department of Medicine, University of Pittsburgh School of Medicine, Pittsburgh, PA 15261, USA. [12]Center for Molecular Medicine Cologne, Cologne, Germany. [13]Cologne Excellence Cluster on Cellular Stress Responses in Aging-Associated Diseases, Cologne, Germany. [14]Department II of Internal Medicine, University Hospital of Cologne, Cologne, Germany. [15]Department of Molecular Medicine, Scripps Research, La Jolla, CA 92037, USA. [16]Institute of Physiology, University Kiel, Kiel, Germany. [17]James A. Haley Veterans' Hospital, Tampa, FL 33612, USA. [18]Hypertension and Kidney Research Center, University of South Florida, Tampa, FL 33602, USA. [19]Present address: Department of Molecular and Medical Pharmacology, David Geffen School of Medicine, University of California, Los Angeles, USA. [20]These authors contributed equally: Markus M. Rinschen, Oleg Palygin, Gary Siuzdak, Alexander Staruschenko. ✉email: rinschen@biomed.au.dk; siuzdak@scripps.edu; staruschenko@usf.edu

