## [Peer Review File · Nature Communications]

Reviewers' Comments:

Reviewer #1:

Remarks to the Author:

Rinschen M.M. et al. aimed to elucidate the association between lysine metabolism and salt-sensitive hypertension. They revealed the excess lysine is an effective treatment of a rat model of salt-sensitive hypertension and the novel metabolite N ϵ -malonyl-lysine correlates with decreased abundance of free fatty acids. They further showed that lysine excretion results in a negative balance on central carbon and carbohydrate metabolism. Similar phenomenons were observed in the rats fed a ketogenic diet.

I think their hypothesis and purpose are intriguing and worth being pursued. Their original metabolomics approach is technically innovative. However, their results are prosaic and not integrated. Although lysine is an essential amino acid and could affect other organs, they only focus on the kidney. I think there are significant issues in this study.

Major issues

- 1) The authors should show the excretion of sodium, body weights, and the level of hormones such as renin and aldosterone when they investigate the lysine-treated hypertensive rats. A previous report showed lysine infusion caused kaliuresis (Am. J. Physiol. 206: 409, 1964). The authors also need to refer to it and to explain the differences from previous reports.
- 2) The data in Fig. 2 showed that lysine induced diuresis after the only 1day of treatment. The authors would need a further investigation about the mechanism of the diuresis such as the release of vasopressin and other hormones.
- 3) It is interesting the lysine infusion ameliorated the kidney injury in D/SS rats. However, the decrease of blood pressure itself could improve the kidney damage. While the authors showed some in vitro studies to investigate the mechanisms, more in vivo studies using another disease animal model should be needed. Thinking about the proteinuria and the fatty acid metabolism, diabetic mice could be a choice. Certification of lysine rich diet effects in other hypertension and CKD models should be needed, such as kidney mass reduction/ 5/6-neohrectomized model. As authors cited, ketogenic diet has been a therapeutic intervention to polycystic kidney disease (PKD, Ref. 41), so how about the a stable isotope labeling strategy with untargeted metabolomic analysis of lysine rich diet or ketogenic diet in PKD model ?
- 4) While the lysine treatment increased albuminuria in Fig. 2, high lysine in the urine was negatively correlated with proteinuria in Fig. 4. Is there any difference between the two conditions?
- 5) Previous reports already showed ketogenic diet could ameliorate kidney injury with weight loss. Although the ketogenic diet could recreate similar conditions to the lysine treatment, it would not be supportive in the context. A lot of metabolic changes could happen not only in the kidney but other organs with lysine treatment and the ketogenic diet. The metabolic changes may be different between lysine treatment and the ketogenic diet. The data shown in Fig. 6 are not enough to convince readers.

Minor issues

- 1) The data in Fig. 1G is not well explained. At least, the authors should indicate what the blue lines and the light blue lines stand for.
- 2) In Fig. 2E, the authors need to show the results of statistical analyses.
- 3) As for Fig. 3C, a more detailed explanation of the figure is needed.
- 4) Quantitative data of the protein plugs and LRP2 would be needed in Fig. 3D.
- 5) Fig. S6F on page 8 does not exist.
- 6) There is no indication about the red and blue rectangles in Fig. 4C and 4E. In Figure4. Authors described "We found that N ϵ -malonyl-lysine was significantly increased in the hypertensive kidneys with lysine (FIG. 4C), ~" (Line 238-239). But I could not distinguish the more significant differences among increasing trends of Ne-malonyl lysine in kidney cortices of hypertensive rat groups (4% salt, 8% salt 2wk, 8% salt 3 wk) compared with that of the control rat group in Fig.4 C. Please make it clear the description.

- 7) The references should be added for 'previous reports' in line 260.
- 8) The authors should show what the abbreviation 'AA' stands for. Also, fructoselysine and saccharopine were not shown in Fig. 5C.
- 9) The Data about 'there was no altered abundance of these metabolites in the urine' in line 266 were not shown.
- 10) As for Fig. S8, each data should be explained in the text.
- 11) The result of statistical analysis should be shown in Fig. 6D.

Reviewer #2:

Remarks to the Author:

Rinschen and coworkers aim to elucidate the renoprotective effects of oral L-lysine loading on the development of kidney disease in hypertensive Dahl salt sensitive rats, a well established animal for salt-induced hypertension and chronic kidney disease. As the authors describe in the introduction the renoprotective effects of L-lysine have been described previously by various groups, and genome-wide GWAS have identified genetic variations in lysine transport via kidney-specific transporter SLC7A9 to be an important modifier of kidney disease.

L-Lysine can be incorporated in protein, with L-lysine residues being important targets for modification through acyl-CoA esters (e.g. acetyl-CoA), can be oxidized through the saccharopine and pipecolate pathways of L-lysine degradation, and can be used as a building block for carnitine and other molecules. The authors applied untargeted metabolomics using ¹³C-labelled L-lysine in mice to elucidate the fate of L-lysine following oral L-lysine load and the mechanism of kidney-protective effects of L-lysine in hypertension.

Although the observation of the renoprotective effects of L-lysine in hypertension has been studied previously, the systematic untargeted metabolomic approach the authors have chosen is somewhat novel in this context.

The study could be considerably improved and the provided arguments strengthened if the following aspects were addressed:

1. L-Lysine metabolism is known to show significant species differences. For example, the enzyme which catalyzes the first step of the pipecolate pathway has not yet been identified in human beings. Therefore, there are still debates about the exact mechanism of pipecolate synthesis in human beings (see Struys et al. *J Inherit Metab* 2014; Posset et al. *J Inherit Metab* 2015). Therefore, the authors should clearly state and discuss that metabolomic studies in mice do not necessarily reflect the lysine metabolome of human beings.
2. The authors used a common ¹³C-labelling strategy. However, since L-lysine can be degraded via α - or ϵ -N-degradation, this methodological approach has some limitations since it does not allow to identify and quantify the exact route of L-lysine degradation. Therefore, it would be important to repeat key experiments using ¹⁵N-labelled L-lysine.
3. L-Lysine competes with L-cystine, L-arginine, and L-ornithine for tubular transport. Therefore, oral L-lysine load has an effect on the tubular transport of the other three amino acids. Since L-arginine is used as substrate for NO metabolism, which is a key regulator of blood pressure, this secondary effect of L-lysine challenge might be an important modifier in the Dahl salt-sensitive rat model. The authors should study and quantify this potentially important mechanism.
4. It is an interesting observation that N- ϵ -malonyl-lysine is increasingly formed during oral L-lysine load. Although this metabolite has not yet been reported, it has been known for some years that protein-bound L-lysine residues are malonylated via the ϵ -N (e.g. Liu et al. *BMC Genomics* 2018). Since the formation of this metabolism seems to be a key finding of this study and since the authors draw various conclusions from this, it would be very important to understand much the underlying mechanism in detail. For instance, it is well known that malonyl-CoA is a key regulator of fatty acid metabolism, being an important substrate of fatty acid synthesis and at the same time an inhibitor of carnitine palmitoyltransferase 1 (CPT1), which is the first step of the carnitine cycle required to transport long-chain fatty acids across the inner mitochondrial

membranes and to make them available for the mitochondrial beta-oxidation of fatty acids. Although the metabolomic effects shown in this study provide some evidence for changes in fatty acid metabolism following L-lysine load, the underlying mechanism remains fairly unclear. It should be tested whether N- ϵ -malonyl-lysine has a direct effect on the enzymatic activities of acetyl-CoA carboxylase, carnitine palmitoyl-CoA transferase 1, and other enzymes of fatty acid synthesis and oxidation. Such mechanism cannot be excluded from the results and, therefore, it does not seem sound to conclude that the demonstrated changes are the result of a lysine-induced reduction of malonyl-CoA and acetyl-CoA.

5. Similarly, the intracellular effects of L-lysine load on intracellular carbon metabolism need some attention. Since acetyl-CoA, glucose, and 2-oxoglutarate (alpha-ketoglutarate), are important energy substrates, the authors should demonstrate whether and to which extent increased formation and excretion of central carbon metabolites affects intracellular energy metabolism (e.g. ATP production) of proximal tubular cells.

6. Saccharopine is formed as an intermediary metabolite from L-lysine in the so-called saccharopine pathway of L-lysine oxidation. The first enzymatic step of the pathway is conducted by alpha-aminoadipic semialdehyde synthase (AASS). AASS is able to catalyze two consecutive enzymatic steps, saccharopine being the product of the first step and the substrate of second step, resulting in the formation of α -amino adipic semialdehyde. Therefore, it requires some explanation why the formation and excretion of saccharopine increases and whether this might reflect a selective inhibition of the second enzymatic step of AASS during L-lysine challenge.

Reviewer #3:

Remarks to the Author:

This is an in-depth analysis of the metabolism of lysine in the kidney and other organs of Dahl salt sensitive rats. The authors describe a heretofore striking anti-hypertensive effect of lysine feeding, and identify a variety of interesting lysine metabolites. While these findings are fascinating and largely new, they seem highly descriptive. It isn't clear from the data presented exactly how lysine lowers blood pressure or really prevents renal injury. A variety of metabolites have been identified, but it isn't clear if they are biologically active. The findings with albuminuria seem opposite of a beneficial effect. Importantly, one begins to wonder if these findings have relevance to other models of hypertension or if these are unique findings to the DSS rat. Ideally, it would be much better if some data in hypertensive humans were available.

There are ample experimental data to implicate changes in vascular and CNS function in hypertension. Does lysine affect either of these?

The data regarding lysine metabolism, degradation and incorporation into other products in figure 1 seem to be from mice. It would be important to compare such results between hypertensive and normotensive animals. For example, between the DSS and DSR rats.

It would seem important to determine if it is lysine or one of its metabolites that reduce blood pressure. If you administer Saccharopine or Fructose lysine, would these lower blood pressure and prevent glomerular damage?

The findings that lysine enhances albuminuria and reduces albumin uptake by proximal tubular cells are interesting but seem opposite of a protective effect. As you point out, albuminuria is deleterious, and yet you show a benefit of lysine administration. Moreover, one wonders if this is an important anti-hypertensive mechanism. A shift in the renal function (pressure natriuresis) curve can occur without overt evidence of renal damage. Such a shift is suggested by the data in figure 2F, and likely underlies the initial rise in blood pressure. Does lysine have any effect on renal sodium transport or the known transporters?

You state that malonyl lysine is increased in hypertensive kidneys with lysine, but if one examines figure 4C, it is clear that there is no major difference between control and salt feeding, suggesting that hypertension really doesn't affect formation of this metabolite.

The lettering and symbols in figure S1A are too small to see.

I could not find the supplemental tables.

REVIEWER COMMENTS

Reviewer #1 (Remarks to the Author):

Rinschen M.M. et al. aimed to elucidate the association between lysine metabolism and salt-sensitive hypertension. They revealed the excess lysine is an effective treatment of a rat model of salt-sensitive hypertension and the novel metabolite N ϵ -malonyl-lysine correlates with decreased abundance of free fatty acids. They further showed that lysine excretion results in a negative balance on central carbon and carbohydrate metabolism. Similar phenomenons were observed in the rats fed a ketogenic diet.

I think their hypothesis and purpose are intriguing and worth being pursued. Their original metabolomics approach is technically innovative. However, their results are prosaic and not integrated. Although lysine is an essential amino acid and could affect other organs, they only focus on the kidney. I think there are significant issues in this study.

Thank you for your comments. We have now added an additional isotope labeling study in the hypertensive rat to address this general point. Using targeted metabolomics, we have shown that lysine metabolism is accelerated in hypertension, chiefly in liver and kidney (new Figure 2). In addition, we also analyzed the effect of hypertension in different organs (now in new Figure 7).

Major issues

1) The authors should show the excretion of sodium, body weights, and the level of hormones such as renin and aldosterone when they investigate the lysine-treated hypertensive rats. A previous report showed lysine infusion caused kaliuresis (Am. J. Physiol. 206: 409, 1964). The authors also need to refer to it and to explain the differences from previous reports.

We measured weight, serum, and urine electrolytes and did not find a significant difference. All of these parameters are summarized as a new supplemental Figure 3 C-F and Table, and additional data are appended below in reviewer Figure 1 and Table 1. While we did not see a difference in urinary potassium, net kaliuresis might be a

contributor to the overall beneficial outcome of lysine in these experiments. We have referenced this study and added kaliuresis as an aspect to the discussion.

Technically and conceptionally, Walkers study from 1964 is different from our study through the following aspects: 1) the administration mode of lysine was intravenous with a sustained infusion of 20-140 $\mu\text{m}/\text{kg min}$, partly together with Mannitol; 2) the context was focused on the regulation of acid-base status and; 3) The model system used was a dog.

Reviewer Figure 1: Total body weight of DSS rats after 2 weeks on HS. No difference between lysine treated (red) and untreated (black) DSS rats.

Blood electrolytes after 2 weeks on HS

	cK	Na	Ca	Cl	Glu	Cre
	mM					
SS HS	3.3±0.2	137±1	1.17±0.02	105±1	345±11	0.37±0.04
SS HS+Lysine	3.2±0.1	138±1	1.26±0.02	109±1	313±13	0.32±0.01

Reviewer Table 1: Blood electrolytes of potassium, Na, Ca, Cl as well as glucose (in μM) and Creatinine. Note the low cK is a typical for DSS rats on HS diet (ref PMID: 33046522).

2) The data in Fig. 2 showed that lysine induced diuresis after the only 1day of treatment. The authors would need a further investigation about the mechanism of the diuresis such as the release of vasopressin and other hormones.

As suggested, we determined copeptin level, a stable parameter of vasopressin (see

Reviewer Figure 2: Copeptin in control and lysine-treated rats on a high salt diet.

Reviewer Figure 2 in this document). We found a tendency to increase with lysine infusion in hypertension at late stages. We think that this could suggest a compensatory effect in the distal nephron.

We also determined further analytes in the serum with relation to the renin-angiotensin-aldosterone system. The data did not show evidence for decreased activity of the RAAS system with Lysine. Data is added in new Table 1, showing partial increase in many bioactive hormones of the RAAS system.

We also checked expression of key transporters such as NKCC2, and NCC, using a broad proteomic approach. We would have expected these transporters to be decreased.

However, our data did not reveal any decreases in these proteins, NKCC2 even appeared to increase. These data are added as a new supplemental Table 4. Nevertheless, the

detailed distal tubular compensation of these models needs to be further elucidated.

In conclusion, we rather think that the increase in diuresis is largely due to 1) increase of GFR and renal blood flow (for instance, described in¹) and 2) osmotic diuresis through lysine. This is added to the discussion.

3) It is interesting the lysine infusion ameliorated the kidney injury in D/SS rats. However, the decrease of blood pressure itself could improve the kidney damage. While the authors showed some in vitro studies to investigate the mechanisms, more in vivo studies using another disease animal model should be needed.

Thinking about the proteinuria and the fatty acid metabolism, diabetic mice could be a choice. Certification of lysine rich diet effects in other hypertension and CKD models should be needed, such as kidney mass reduction/ 5/6-neohrectomized model. As authors cited, ketogenic diet has been a therapeutic intervention to polycystic kidney disease (PKD, Ref. 41), so how about the a stable isotope labeling strategy with untargeted metabolomic analysis of lysine rich diet or ketogenic diet in PKD model ?

Thank you for these excellent suggestions. Your comment comes down to the causal connection between lysine metabolism, kidney damage, and hypertension. Therefore, we first tested whether lysine supplementation alone reduced hypertension in a model of spontaneous hypertension, the SHR rat. This rat does not develop kidney disease but spontaneous hypertension. The analysis revealed that lysine alone does not alter hypertension.

All your other suggestions are valid as well but do not match our hypothesis. Our previous results showed that lysine metabolism is altered in the tubules of the Dahl Salt Sensitive rat model, thereby leading to the current study. The 5/6 nephrectomy rats do not have alterations in lysine metabolism in any organ². Cystic kidney disease has not been shown to be driven by large alteration in lysine modification or Na-mediated lysine degradation^{3,4}. The cystic kidney disease model is also not presenting with hypertension in our hands. Thus, we would not expect that either of these models would have the opportunity to exploit lysine metabolism.

By stating this, we do not wish to make any excuses. We subjected rats with cystic kidney disease (PCK) to the same high lysine diet and found, as expected, no effect of lysine on bodyweight and cyst formation. In addition, diuresis did not change in early or late stages of the disease. These data are appended as the reviewer Figure 3 next to this paragraph. We found a difference in diuresis only at 14 weeks old animals (between treated and non-treated). We also analyzed the electrolytes in body fluids and did not see any differences.

Instead, we thought that the true impact would be to translate this naturally occurring model of salt-sensitive of hypertension to humans. Would acute lysine challenge recapitulate the same physiology in humans? To this end, we challenged humans in a

pilot trial with a large peroral load of lysine, followed by 12 hrs urine collection. Humans with reduced nephron mass (risk for hypertension) have alterations similar to the metabolic alterations observed in rats with kidney damage and hypertension. In addition, the data revealed similar metabolic capacities, including increased lysine modification and reduced TCA excretion, a pattern similar to the protective effect in lysine rats. We have added these new data a Figure 8.

4) While the lysine treatment increased albuminuria in Fig. 2, high lysine in the urine was negatively correlated with proteinuria in Fig. 4. Is there any difference between the two conditions?

We think that the reviewer referred to Figure 3, now Figure 4. Acutely, lysine binds to Megalin-Cubilin, prevents albumin reabsorption in PT and releases an excessively filtered albumin accumulated in the kidney (Figure 4). Administered over a longer time period, lysine improves kidney metabolism by ameliorating injury and reducing fibrosis (Figure 3,5,6). Induction of albuminuria can be seen only at very early time points, as in Figure 4C, and consistent with literature from the physiology field⁵. We have now marked this also in the figure and clarified the figure legends.

5) Previous reports already showed ketogenic diet could ameliorate kidney injury with weight loss. Although the ketogenic diet could recreate similar conditions to the lysine treatment, it would not be supportive in the context. A lot of metabolic changes could happen not only in the kidney but other organs with lysine treatment and the ketogenic diet. The metabolic changes may be different between lysine treatment and the ketogenic diet. The data shown in Fig. 6 are not enough to convince readers.

We appreciate the feedback that Figure 6 is not supportive. Given the fact that additional studies in the cystic kidney disease model showed no benefit of lysine supplementation in kidney disease, we have refocused the manuscript by removing these data. Instead, we have added further data on a spontaneous hypertensive rat and humans with kidney disease as discussed above.

Minor issues

1) The data in Fig. 1G is not well explained. At least, the authors should indicate what the blue lines and the light blue lines stand for. **We clarified this. The dark lines are proteins, the light blue lines are metabolites.**

2) In Fig. 2E, the authors need to show the results of statistical analyses. **This is now replaced by a larger overview of 15N tracing.**

3) As for Fig. 3C, a more detailed explanation of the figure is needed. **We have added more details.**

4) Quantitative data of the protein plugs and LRP2 would be needed in Fig. 3D. **We have done this quantification for the plugs – see reviewer Figure 4. We rephrased the statement regarding the LRP2 – here we want to focus on the more “patchy” staining**

5) Fig. S6F on page 8 does not exist. **Corrected – was referring to Fig. S6C.**

6) There is no indication about the red and blue rectangles in Fig. 4C and 4E. **Thank you, this was fixed.**

In Figure 4. Authors described “We found that Ne-malonyl-lysine was significantly increased in the hypertensive kidneys with lysine (FIG. 4C), ~” (Line 238-239). But I could not distinguish the more significant differences among increasing trends of Ne-malonyl lysine in kidney cortices of hypertensive rat groups (4% salt, 8% salt 2wk, 8% salt 3 wk) compared with that of the control rat group in Fig. 4 C. Please make it clear the description.

Thank you, we have clarified this. We have added additional data that shows that Ne-malonyl-lysine formation was further accelerated.

7) The references should be added for ‘previous reports’ in line 260. **We added the reference.**

8) The authors should show what the abbreviation ‘AA’ stands for. Also, fructoselysine and saccharopine were not shown in Fig. 5C. **Thank you, we have revised this. AA is an abbreviation for amino acids, we have added this.**

9) The Data about ‘there was no altered abundance of these metabolites in the urine’ in line 266 were not shown. **We have deleted this statement.**

10) As for Fig. S8, each data should be explained in the text. **Thank you, this is now removed.**

11) The result of statistical analysis should be shown in Fig. 6D. **This is now replaced.**

Reviewer #2 (Remarks to the Author):

Rinschen and coworkers aim to elucidate the renoprotective effects of oral L-lysine loading on the development of kidney disease in hypertensive Dahl salt sensitive rats, a well established animal for salt-induced hypertension and chronic kidney disease. As the authors describe in the introduction the renoprotective effects of L-lysine have been described previously by various groups, and genome-wide GWAS have identified genetic variations in lysine transport via kidney-specific transporter SLC7A9 to be an important modifier of kidney disease.

L-Lysine can be incorporated in protein, with L-lysine residues being important targets for modification through acyl-CoA esters (e.g. acetyl-CoA), can be oxidized through the saccharopine and pipecolate pathways of L-lysine degradation, and can be used as a building block for carnitine and other molecules. The authors applied untargeted metabolomics using ¹³C-labelled L-lysine in mice to elucidate the fate of L-lysine following oral L-lysine load and the mechanism of kidney-protective effects of L-lysine in hypertension.

Although the observation of the renoprotective effects of L-lysine in hypertension has been

studied previously, the systematic untargeted metabolomic approach the authors have chosen is somewhat novel in this context.

We thank the reviewer for acknowledging the novelty of the manuscript. In fact, while the protective role has been implicated in several large-scale studies, there have never been mechanistic studies of lysine in disease. Thus, this study is the first to directly show lysine's beneficial effect and mechanism in kidney protection.

The study could be considerably improved and the provided arguments strengthened if the following aspects were addressed:

1. L-Lysine metabolism is known to show significant species differences. For example, the enzyme which catalyzes the first step of the pipecolate pathway has not yet been identified in human beings. Therefore, there are still debates about the exact mechanism of pipecolate synthesis in human beings (see Struys et al. J Inherit Metab 2014; Posset et al. J Inherit Metab 2015). Therefore, the authors should clearly state and discuss that metabolomic studies in mice do not necessarily reflect the lysine metabolome of human beings.

Point well taken and acknowledged. We have mentioned this and other limitations in the discussion section.

To further address human relevance of our findings, we subjected patients with kidney disease to a one-time large lysine challenge (New data added as Figure 8). Note that in terms of lysine degradation and modification in response to lysine challenge, humans appear to react similar to rats.

2. The authors used a common ^{13}C -labelling strategy. However, since L-lysine can be degraded via α - or ϵ -N-degradation, this methodological approach has some limitations since it does not allow to identify and quantify the exact route of L-lysine degradation. Therefore, it would be important to repeat key experiments using ^{15}N -labelled L-lysine.

We have repeated the isotope labeling in vivo using ^{15}N - labeling (Now added in Figure 2; See also reviewer Figure 5 below). To further corroborate the direct metabolic capabilities of the kidney, we isolated proximal tubules and incubated them with both ^{15}N - labeled lysine. We found that the Na ^{15}N lysine was found in the degradation product amino adipate. Pipecolic acid was not detected in the kidney tubules. Thus, we conclude that the saccaropine pathway (and Na-degradation) is the most frequent and predominant pathway in the kidney, consistent with our previous data.

Reviewer Figure 5: ¹⁵N-lysine metabolism in isolated proximal tubules. A. Biochemical pathway of N ϵ and N α degradation. B. Treatment scheme of a living proximal tubule suspension. C. relative increase in ¹⁵N incorporation in Saccaropine. D relative increase in ¹⁵N incorporation in amino adipic acid (AAA). No alterations are observed. E. Increase in unlabeled AAA, suggesting that lysine is degraded via N α -degradation. Pipecolic acid was not observed in isolated tubules.

3. L-Lysine competes with L-cystine, L-arginine, and L-ornithine for tubular transport. Therefore, oral L-lysine load has an effect on the tubular transport of the other three amino acids. Since L-arginine is used as substrate for NO metabolism, which is a key regulator of blood pressure, this secondary effect of L-lysine challenge might be an important modifier in the Dahl salt-sensitive rat model. The authors should study and quantify this potentially important mechanism.

The reviewer is right. To analyze whether this mechanism is of relevance in vivo,

1. We reanalyzed urine from lysine treated rats using targeted metabolomics. We did not see any changes in the three metabolites, consistent with the initial untargeted metabolomics analysis presented in Figure 6.
2. To investigate if this has a systemic impact, we analyzed also the abundance of metabolites in different organs. We did not find – with the exception of decreased ornithine abundance in heart – an effect of lysine treatment on the abundance of these metabolites (reviewer figure 6 below).
3. Furthermore, we analyzed the concentration of these metabolites in urines of human challenged with lysine (healthy and with kidney disease). Here, we found that urinary arginine had a large variation, but trended to increase with lysine administrations (Supplementary Figure 8, Figure 8).
4. We analyzed the dynamic uptake of ¹³C labeled arginine ex vivo in the presence of different lysine concentrations and found only minimal effects on arginine, and, generation of citrullin and ornithine.

This being said, this mechanism is possibly relevant, and might require more physiologically resolved studies as compared to bulk urine and bulk tissue data. We have added this point to the limitations of the paper.

Reviewer Figure 6: Arginine, Ornithine and Cystine signals in various organs of DSS treated. Ornithine intensity was reduced in Muscle, and significantly decreased in heart.

4. It is an interesting observation that N-ε-malonyl-lysine is increasingly formed during oral L-lysine load. Although this metabolite has not yet been reported, it has been known for some years that protein-bound L-lysine residues are malonylated via the ε-N (e.g. Liu et al. BMC Genomics 2018). Since the formation of this metabolism seems to be a key finding of this study and since the authors draw various conclusions from this, it would be very important to understand much the underlying mechanism in detail. For instance, it is well known that malonyl-CoA is a key regulator of fatty acid metabolism, being an important substrate of fatty acid synthesis and at the same time an inhibitor of carnitine palmitoyltransferase 1 (CPT1), which is the first step of the carnitine cycle required to transport long-chain fatty acids across the inner mitochondrial membranes and to make them available for the mitochondrial beta-oxidation of fatty acids. Although the metabolomic effects shown in this study provide some evidence for changes in fatty acid metabolism following L-lysine load, the underlying mechanism remains fairly unclear. It should be at tested whether N-ε-malonyl-lysine has a direct effect on the enzymatic activities of acetyl-CoA carboxylase, carnitine palmitoyl-CoA transferase 1, and other enzymes of fatty acid synthesis and oxidation. Such mechanism cannot be excluded from the results and, therefore, it does not seem sound to conclude that the demonstrated changes are the result of a lysine-induced reduction of malonyl-CoA and acetyl-CoA.

We appreciate the suggestion. We tested, as suggested, the activity of Ne-malonyl-lysine and Na-malonyl-lysine on the two key enzymes of fatty acid synthesis, including ACC activity, FASN activity and beta oxidation.

1. Using an in vitro FASN assay using bovine recombinant protein, we observed no alterations with Ne-Malonyllysine. However, we observed a 20%, significant inhibition of FASN activity with unphysiologically high concentrations of alpha-Malonyl-lysine (Reviewer Figure 7). Since the compound N-a-Malonyl-lysine could only be synthesized with lower purity (80%, as opposed to N-e-Malonyl-lysine >99%), and was also present at lower intensities, we prefer not to include these data in the manuscript.
2. Regarding ACC activity, we measured activity with control and Ne-Malonyl-lysine spike-in in cell culture lysate using a commercial assay from MyBiosource. We observed a very small

Reviewer Figure 7. FASN activity.

~15% decrease (n=8) in ACC activity, but only with a very high abundance of Ne-malonyl-lysine (c= 1 mM). Because of the rather high concentration and the questionable relevance of proximal tubule cell cultures, we have included these data in the Supplemental Figure.

3. Regarding beta-oxidation, we found an increased abundance of beta oxidation enzymes in proteomic analysis of lysine-treated rats. However, functionally, we did not see a global alteration of acylcarnitines, as expected with altered beta-oxidation activity. These data are now available in Supplemental Table 4. This could be a mechanism in vivo that may be directly or indirectly related to malonyl-lysine.

In addition, we checked the formation of the metabolite in vivo using isotope tracing studies. We found that the metabolite formation was accelerated in hypertension. This supports, in our view, the original hypothesis: formation of Ne-malonyl-lysine is a byproduct of lysine excess that leads to an overall depletion of Malonyl-CoA in the kidney. This accelerated lysine conjugation appears to be, as also shown through more global data in Figure 2, a general feature of hypertension.

5. Similarly, the intracellular effects of L-lysine load on intracellular carbon metabolism need some attention. Since acetyl-CoA, glucose, and 2-oxoglutarate (alpha-ketoglutarate), are important energy substrates, the authors should demonstrate whether and to which extent increased formation and excretion of central carbon metabolites affects intracellular energy metabolism (e.g. ATP production) of proximal tubular cells.

This is a fair point. It is long known that proximal tubules can directly use a variety of substrates for energy metabolism, including pyruvate, acetate, lactate, alpha-Ketoglutarate, amino acids, and albumin-bound free fatty acids. Analysis of metabolomics data revealed a decreased abundance of AMP with lysine after 2 weeks with 8% hypertension (log₂ ratio -0.81) and increased after 3 weeks (log₂ ratio 1.2) – shown in the supplemental data of the paper.

To further model this, we went back to the in vitro model (OK cells). Unfortunately, these cells do not mimic the complete metabolic physiology but allow for modeling essential proximal tubule functions, such as albumin endocytosis. We asked whether albumin overload had an effect on metabolism, and if lysine could block this. We observed a decrease in ATP production with albumin exposure, which could be reversed with lysine treatment, suggesting a protective role of lysine on PTC. These findings are consistent with the early time point of the metabolomics data (see reviewer figure 9 below).

Thus, we think that inhibition of albumin uptake could also contribute to the protective energetic effect. However, further analyses are necessary to dissect how fuel availability governs the proximal tubules adaptation in hypertension.

Reviewer Figure 8. *The Agilent Seahorse Mitochondria Stress Test revealed the protective effect of Lysine on mitochondrial respiration in cultured proximal tubule cells exposed to albumin.* A. Seahorse XF Cell Mito Stress Test shows that cultured proximal tubule cells treated with BSA (4mg/ml; 3 hours exposure) display diminished basal, spare respiratory capacity, maximal respiration and ATP production compared to cells treated with Lysine (1,10, and 50 mM, respectively). B. Test assay design and standard output parameters. C. Statistical summary for Basal and Spare Respiratory Capacity (n=6, ANOVA, Tukey post hoc comparison, $p < 0.05$). D. Statistical summary for Proton Leak and ATP Production (n=6, ANOVA, Tukey post hoc comparison, $p < 0.05$). OCR, oxygen consumption rate.

6. Saccharopine is formed as an intermediary metabolite from L-lysine in the so-called saccharopine pathway of L-lysine oxidation. The first enzymatic step of the pathway is conducted by alpha-amino adipic semialdehyde synthase (AASS). AASS is able to catalyze two consecutive enzymatic steps, saccharopine being the product of the first step and the substrate of the second step, resulting in the formation of α -amino adipic semialdehyde. Therefore, it requires some explanation why the formation and excretion of saccharopine increases and whether this might reflect a selective inhibition of the second enzymatic step of AASS during L-lysine challenge.

Great point. First, we wanted to clarify whether Alllysine (or α -amino adipic semialdehyde) was increased through lysine. Alllysine was not detected by the untargeted metabolomics analysis, since it fell under the threshold for reliable quantification. Thus, we have repeated the analysis using a more sensitive (triple-quad) instrument. The data reveals an increase also in Alllysine, suggesting that both abundances in Alllysine and Saccharopine are increased.

To explain the increased formation of saccharopine, we performed proteomic analysis. We found that the expression of AASS was significantly increased in all three hypertensive models through lysine, but not in the control models. Interestingly, the subsequent enzymes were not regulated as strongly. The data is now included in the supplemental Figure 5, and also shown below. Thus, we feel that increased expression of the enzyme in the presence of a large amount of lysine may be partially responsible for the large increase of saccharopine.

Reviewer Figure 9: Proteomic analysis of enzymes of lysine degradation in the kidney cortex of different dietary interventions in the hypertensive DSS disease model.

Reviewer #3 (Remarks to the Author):

This is an in-depth analysis of the metabolism of lysine in the kidney and other organs of Dahl salt sensitive rats. The authors describe a here to fore striking anti-hypertensive effect of lysine feeding, and identify a variety of interesting lysine metabolites. While these findings are fascinating and largely new, they seem highly descriptive. It isn't clear from the data presented exactly how lysine lowers blood pressure or really prevents renal injury. A variety of metabolites have been identified, but it isn't clear if they are biologically active. The findings with albuminuria seem opposite of a beneficial effect.

The reviewer raises an important point at the center of understanding complex diseases such as hypertensive organ damage. In most cases, there is not a single trigger for the disease, and also not a single mechanism that can be attributed to the kidney-protective effect of lysine. Our goal was to investigate the entity of lysine metabolites. Our conclusion is that lysine and its metabolite exert a pleiotropic effect that conveys protection in hypertension and kidney disease. This concept of "activity metabolomics" takes a metabolite-centric perspective to alter complex physiology through various mechanisms.

Regarding the albuminuria findings: Primarily, lysine blocks albumin uptake in the proximal tubule. This leads to more albuminuria in the short term, which is representing the release of an excessive albumin accumulation in the kidney nephron (Figure 3A). However, chronic supplementation of lysine leading to decrease in albuminuria and renal damage (Figure 2) through protection of both proximal tubules and glomerular function. The blockage of albumin uptake could be a positive effect when more glomerular damage is present, leading to an overall increase in tubular function and the reduction in oxidative stress.

Consistent with this concept, in vitro data in OK cells show a connection between lysine treatment and albumin-induced oxidative stress. Please see reviewer only Figure 7.

Importantly, one begins to wonder if these findings have relevance to other models of hypertension or if these are unique findings to the DSS rat. Ideally, it would be much better if some data in hypertensive humans were available.

We recapitulated the experiment of lysine challenge in humans as suggested in a pilot study (new Figure 8). We analyzed healthy volunteers and patients at risk for hypertensive kidney disease (i.e., hypertensive patients, either a solitary kidney or mild proteinuria). The metabolic and tubular proteinuric effects within both groups were consistent with the findings in the rat model. Please note that the lysine intervention was only done for a day, and a more extended clinical trial was not within the scope of a revision. This is something we would like to follow up in the future.

There are ample experimental data to implicate changes in vascular and CNS function in hypertension. Does lysine affect either of these?

Vascular damage is mediating hypertension in the SHR rat. This rat is not affected by lysine treatment. These new data are now included in the figure 7.

A central mechanism regulating lysine is through the renin angiotensin aldosterone system. We did not see changes consistent with an antihypertensive effect. These data are added to the manuscript as a new Table 1. The findings cannot explain the antihypertensive effects we observed.

Our labs do not have the expertise to analyze CNS function. However, we profiled the lysine metabolome in several organs. Interestingly, while many organs such as heart, muscle, kidney or liver react in response to a stimulus with high lysine with lysine modification, the brain does not (see reviewer figure 10 below).

Reviewer Figure 10. Results of intensity of lysine modification metabolites in hypertensive DSS rats in different organs. Saccharopine and acetyllysine increased in kidney and liver, but not in brain.

The data regarding lysine metabolism, degradation and incorporation into other products in figure 1 seem to be from mice. It would be important to compare such results between hypertensive and normotensive animals. For example, between the DSS and DSR rats.

Yes. We have done that. Please note that it is currently not possible to do an identical feeding study with isotope labeled lysine in rats. This is because of the high price of the chow. Rat chows are also not commercially available.

Instead, we injected hypertensive and non-hypertensive DSS rats with isotope labeled lysine (Data described in a new Figure 2). The results show an acceleration of lysine metabolism in hypertensive DSS rat as opposed to a DSS rat without hypertension.

We have further analyzed the effect in spontaneous hypertensive rats (SHR). These rats did not respond with beneficial effects in hypertension and proteinuria. These data are added as a new Figure 7.

It would seem important to determine if it is lysine or one of its metabolites that reduce blood pressure. If you administer Saccharophine or Fructose lysine, would these lower blood pressure and prevent glomerular damage?

We unfortunately are not able to provide an answer to this question. Saccharopine and Fructose-lysine are not commonly available compounds beyond small amounts as analytical standards. A few mg of these compounds consistently ranged between 300 and 500 USD from different, uncommon vendors– and such an feeding analysis would require grams of compound. Despite our efforts we were not able to find a company or a chemist who could perform an accurate synthesis for these molecules in the range of miligrams or even grams at a reasonable purity so we could do a relevant intervention study. Further in vitro analyses we have done regarding metabolic readouts of lysine function remained negative. Thus, we cannot answer the question at this time point, but we have acknowledged this limitation and hope that this paper will increase the interest in these poorly studied, but abundantly formed metabolites.

The findings that lysine enhances albuminuria and reduces albumin uptake by proximal tubular cells are interesting but seem opposite of a protective effect. As you point out, albuminuria is deleterious, and yet you show a benefit of lysine administration.

We think that a prevention of albumin overloading of the proximal tubule is part of the protective mechanism. As seen in Figure 3, the initial response to lysine is an increase in proteinuria. However, with further continuation of the lysine regimen, a protective effect can be seen that again shows a decrease of proteinuria (Figure 2). The associated mechanism also involves significant metabolic alterations (Figure 8).

It is worth noting that patients with CUBLN mutation, leading to a low-molecular weight proteinuria through a similar mechanism of inhibition of albumin uptake, has no overt phenotype in humans⁶. This paper is now cited in the discussion as well.

Moreover, one wonders if this is an important anti-hypertensive mechanism. A shift in the renal function (pressure natriuresis) curve can occur without over evidence of renal damage. Such a shift is suggested by the data in figure 2F, and likely underlies the initial rise in blood pressure. Does lysine have any effect on renal sodium transport or the known transporters?

We performed analysis of the known kidney salt transporters by proteomics. We did not find a decrease in abundance of any of the transporters, including NCC and NKCC2.

We also performed further characterization of the urinary sodium and potassium excretion, as well as the hormone status of the RAAS system. These new data are added as Table 1 and Supplemental Figure 3. The regulation observed here are not consistent with a central effect of lysine on hypertension.

We also analyzed the effect of lysine treatment on the expression of renal proximal tubule transporters as well as other transporters involved in salt transport. Here, we found that NCC and NKCC2 transporters trended to even increase. However, lysine decreased abundance of several proximal tubule sodium co-transporters. The proteomic analysis results are given now in supplemental Figure 5, and the dataset is added to the paper in supplemental Figure 4.

You state that malonyl lysine is increased in hypertensive kidneys with lysine, but if one examines figure 4C, it is clear that there is no major difference between control and salt feeding, suggesting that hypertension really doesn't affect formation of this metabolite.

Yes and no. The abundance is not altered, but metabolic fluxes can change independent of the abundance of the metabolite. We tested this. In fact, ML-s formation speed (as observed through the abundance of isotope labeled malonyl-lysine after injection of isotope labeled lysine) is increased in hypertension with lysine. This could be responsible for the accelerated depletion of Malonyl-CoA. This is now added as a new panel in a new Figure 5 of the paper.

The lettering and symbols in figure S1A are too small to see.

I could not find the supplemental tables.

We have fixed these issues.

References

1. Deng, A. *et al.* Regulation of oxygen utilization by angiotensin II in chronic kidney disease. *Kidney Int.* **75**, 197–204 (2009).
2. Hanifa, M. A. *et al.* Tissue, urine and blood metabolite signatures of chronic kidney disease in the 5/6 nephrectomy rat model. *Metabolomics* **15**, 112 (2019).
3. Taylor, S. L. *et al.* A metabolomics approach using juvenile cystic mice to identify urinary biomarkers and altered pathways in polycystic kidney disease. *Am. J. Physiol.-Ren. Physiol.* **298**, F909–F922 (2010).

4. Menezes, L. F., Lin, C.-C., Zhou, F. & Germino, G. G. Fatty Acid Oxidation is Impaired in An Orthologous Mouse Model of Autosomal Dominant Polycystic Kidney Disease. *EBioMedicine* **5**, 183–192 (2016).
5. Thelle, K., Christensen, E. I., Vorum, H., Ørskov, H. & Birn, H. Characterization of proteinuria and tubular protein uptake in a new model of oral L-lysine administration in rats. *Kidney Int.* **69**, 1333–1340 (2006).
6. Bedin, M. *et al.* Human C-terminal CUBN variants associate with chronic proteinuria and normal renal function. *J. Clin. Invest.* **130**, 335–344 (2020).

Reviewers' Comments:

Reviewer #1:

Remarks to the Author:

Journal : Nat. Commun. Peer Review

Authors : Markus M. Rinschen, Oleg Palygin, Daria Golosova, Xavier Domingo-Almenara, Amelia Palermo, Michael A. Schafroth⁷, Carlos Guijas¹, Megan L. Gliozzi, Jingchuan Xue , Martin Hoehne, Thomas Benzing, Bernard P. Kok, Enrique Saez, Markus Bleich, Nina Himmerkus, Ora A. Weisz, Benjamin F. Cravatt, Marcus Krueger, H. Paul Benton, Gary Siuzdak and Alexander Staruschenko

Title : Lysine metabolism conveys kidney protection in hypertension.

Manuscript ID : NCOMMS-20-45521

Comment to Revised NCOMMS-20-45521

In this revised paper, Rinschen M.M. et al. added an additional isotope labeling study in the hypertensive rat to address that lysine metabolism is accelerated in hypertension, chiefly in liver and kidney (new Figure 2) and analyzed the effect of hypertension in different organs (now in new Figure 7).

Major issues

1) The authors should show the excretion of sodium, body weights, and the level of hormones such as renin and aldosterone when they investigate the lysine-treated hypertensive rats. Authors measured weight, serum, and urine electrolytes and did not find a significant Difference (supplemental Figure 3 C-F and Table)

A previous report showed lysine infusion caused kaliuresis (Am. J. Physiol. 206: 409, 1964). The authors also need to refer to it and to explain the differences from previous reports.

Authors explain the different aspects of Walkers study from 1964 as following

: 1) the administration mode of lysine was intravenous with a sustained infusion of 20-140 $\mu\text{g}/\text{kg}$ min, partly together with Mannitol; 2) the context was focused on the regulation of acid-base status and; 3) The model system used was a dog.

2) The data in Fig. 2 showed that lysine induced diuresis after the only 1day of treatment. The authors would need a further investigation about the mechanism of the diuresis such as the release of vasopressin and other hormones.

Authors determined copeptin level, a stable parameter of vasopressin (Figure 2 in Authors replay). They also determined further analytes in the serum with relation to the renin-angiotensin-aldosterone system (Table1).

They added the data of expression of key transporters such as NKCC2, and NCC, using a broad proteomic approach and showed that NKCC2 appeared to increase (supplemental Table 4). They concluded that the increase in diuresis is largely due to 1) increase of GFR and renal blood flow (for instance, described in1) and 2) osmotic diuresis through lysine. This is added to the discussion.

3) It is interesting the lysine infusion ameliorated the kidney injury in D/SS rats. However, the decrease of blood pressure itself could improve the kidney damage. While the authors showed some in vitro studies to investigate the mechanisms, more in vivo studies using another disease animal model should be needed.

Thinking about the proteinuria and the fatty acid metabolism, diabetic mice could be a choice. Certification of lysine rich diet effects in other hypertension and CKD models should be needed, such as kidney mass reduction/ 5/6-neohrectomized model. As authors cited, ketogenic diet has been a therapeutic intervention to polycystic kidney disease (PKD, Ref. 41), so how about the a stable isotope labeling strategy with untargeted metabolomic analysis of lysine rich diet or ketogenic diet in PKD model ?

Authors replied that they first tested whether lysine supplementation alone reduced hypertension in a model of spontaneous hypertension, the SHR rat. This rat does not develop kidney disease but spontaneous hypertension. The analysis revealed that lysine alone does not alter hypertension (New Figure 7).

They stated that their previous results showed that lysine metabolism is altered in the tubules of the Dahl Salt Sensitive rat model, thereby leading to the current study. They explained that the 5/6 nephrectomy rats do not have alterations in lysine metabolism in any organ.

Ref 2. Hanifa, M. A. et al. Tissue, urine and blood metabolite signatures of chronic kidney disease in the 5/6 nephrectomy rat model. *Metabolomics* 15, 112 (2019).

They referenced two reports for explaining that cystic kidney disease has not been shown to be driven by large alteration in lysine modification or Na-mediated lysine degradation.

Ref 3) Taylor, S. L. et al. A metabolomics approach using juvenile cystic mice to identify urinary biomarkers and altered pathways in polycystic kidney disease. *Am. J. Physiol.-Ren. Physiol.* 298, F909–F922 (2010).

Ref 4.) Menezes, L. F., Lin, C.-C., Zhou, F. & Germino, G. G. Fatty Acid Oxidation is Impaired in An Orthologous Mouse Model of Autosomal Dominant Polycystic Kidney Disease. *EBioMedicine* 5, 183–192 (2016)..

They subjected rats with cystic kidney disease (PCK) to the same high lysine diet and found, as expected, no effect of lysine on bodyweight and cyst formation. In addition, diuresis did not change in early or late stages of the disease (the reviewer Figure 3).

Finally, authors challenged humans in a pilot trial with a large peroral load of lysine, followed by 12 hrs urine collection.

Humans with reduced nephron mass (risk for hypertension) have alterations similar to the metabolic alterations observed in rats with kidney damage and hypertension. In addition, the data revealed similar metabolic capacities, including increased lysine modification and reduced TCA excretion, a pattern similar to the protective effect in lysine rats (new Figure 8).

4) While the lysine treatment increased albuminuria in Fig. 2, high lysine in the urine was negatively correlated with proteinuria in Fig. 4. Is there any difference between the two conditions?

Authors explained that in acute phase, lysine binds to Megalin-Cubilin, prevents albumin reabsorption in PT and releases an excessively filtered albumin accumulated in the kidney (Figure 4). Induction of albuminuria can be seen only at very early time points and in longer lysine administered time period, lysine improves kidney metabolism by ameliorating injury and reducing fibrosis (Figure 3,5,6).

5) Previous reports already showed ketogenic diet could ameliorate kidney injury with weight loss. Although the ketogenic diet could recreate similar conditions to the lysine treatment, it would not be supportive in the context. A lot of metabolic changes could happen not only in the kidney but other organs with lysine treatment and the ketogenic diet. The metabolic changes may be different between lysine treatment and the ketogenic diet. The data shown in Fig. 6 are not enough to convince readers.

They refocused the manuscript by removing these data (previous manuscript Figure 6) and added further data on a spontaneous hypertensive rat and humans with kidney disease (new Figure 7).

Minor issues are properly corrected

Reviewer #2:

Remarks to the Author:

The authors have addressed all major and minor points of criticism in detail and included significant changes into the revised manuscript, convincingly providing new experimental data to confirm their hypothesis. Furthermore, they addressed potential limitations of their study appropriately.

Although kidney damage is the main focus, possible changes due to long-term lysine intake in other organs still need a more detailed analysis before studying long-term lysine intake in patients.

Rinschen and coworkers have already started first investigations, showing that heart, muscle and liver respond to lysine, but not the brain. This difference should be explained in appropriate detail in future experiments.

RESPONSE TO REVIEWERS' COMMENTS

We thank the editors and reviewers for the careful review of our manuscript. We have not made any changes because none were suggested.

Reviewer #1 (Remarks to the Author):

Journal : Nat. Commun. Peer Review

Authors : Markus M. Rinschen, Oleg Palygin, Daria Golosova, Xavier Domingo-Almenara, Amelia Palermo, Michael A. Schafroth⁷, Carlos Guijas¹, Megan L. Gliozzi, Jingchuan Xue , Martin Hoehne, Thomas Benzing, Bernard P. Kok, Enrique Saez, Markus Bleich, Nina Himmerkus, Ora A. Weisz, Benjamin F. Cravatt, Marcus Krueger, H. Paul Benton, Gary Siuzdak and Alexander Staruschenko

Title : Lysine metabolism conveys kidney protection in hypertension.

Manuscript ID : NCOMMS-20-45521

Comment to Revised NCOMMS-20-45521

In this revised paper, Rinschen M.M. et al. added an additional isotope labeling study in the hypertensive rat to address that lysine metabolism is accelerated in hypertension, chiefly in liver and kidney (new Figure 2) and analyzed the effect of hypertension in different organs (now in new Figure 7).

We thank the reviewer for summarizing our responses and paraphrasing our extensive revisions while bringing up no new points of criticism.

Major issues

1) The authors should show the excretion of sodium, body weights, and the level of hormones such as renin and aldosterone when they investigate the lysine-treated hypertensive rats.

Authors measured weight, serum, and urine electrolytes and did not find a significant Difference (supplemental Figure 3 C-F and Table)

Yes, correct.

A previous report showed lysine infusion caused kaliuresis (Am. J. Physiol. 206: 409, 1964). The authors also need to refer to it and to explain the differences from previous reports.

Authors explain the different aspects of Walker's study from 1964 as following

: 1) the administration mode of lysine was intravenous with a sustained infusion of 20-140 $\mu\text{g}/\text{kg}$ min, partly together with Mannitol; 2) the context was focused on the regulation of acid-base status and; 3) The model system used was a dog.

Yes, correct.

2) The data in Fig. 2 showed that lysine induced diuresis after the only 1 day of treatment. The authors would need a further investigation about the mechanism of the diuresis such as the release of vasopressin and other hormones.

Authors determined copeptin level, a stable parameter of vasopressin (Figure 2 in Authors replay). They also determined further analytes in the serum with relation to the renin-angiotensin-aldosterone system (Table1).

They added the data of expression of key transporters such as NKCC2, and NCC, using a broad proteomic approach and showed that NKCC2 appeared to increase (supplemental Table 4). They concluded that the increase in diuresis is largely due to 1) increase of GFR and renal blood flow (for instance, described in1) and 2) osmotic diuresis through lysine. This is added to the discussion.

Yes, correct.

3) It is interesting the lysine infusion ameliorated the kidney injury in D/SS rats. However, the decrease of blood pressure itself could improve the kidney damage. While the authors showed some in vitro studies to investigate the mechanisms, more in vivo studies using another disease animal model should be needed.

Thinking about the proteinuria and the fatty acid metabolism, diabetic mice could be a choice. Certification of lysine rich diet effects in other hypertension and CKD models should be needed, such as kidney mass reduction/ 5/6-nephrectomized model. As authors cited, ketogenic diet has been a therapeutic intervention to polycystic kidney disease (PKD, Ref. 41), so how about the a stable isotope labeling strategy with untargeted metabolomic analysis of lysine rich diet or ketogenic diet in PKD model ?

Authors replied that they first tested whether lysine supplementation alone reduced hypertension in a model of spontaneous hypertension, the SHR rat. This rat does not develop kidney disease but spontaneous hypertension. The analysis revealed that lysine alone does not alter hypertension (New Figure 7).

Yes, correct.

They stated that their previous results showed that lysine metabolism is altered in the tubules of the Dahl Salt Sensitive rat model, thereby leading to the current study. They explained that the 5/6 nephrectomy rats do not have alterations in lysine metabolism in any organ.

Ref 2. Hanifa, M. A. et al. Tissue, urine and blood metabolite signatures of chronic kidney disease in the 5/6 nephrectomy rat model. *Metabolomics* 15, 112 (2019).

They referenced two reports for explaining that cystic kidney disease has not been shown to be driven by large alteration in lysine modification or Na-mediated lysine degradation.

Ref 3) Taylor, S. L. et al. A metabolomics approach using juvenile cystic mice to identify urinary biomarkers and altered pathways in polycystic kidney disease. *Am. J. Physiol.-Ren. Physiol.* 298, F909–F922 (2010).

Ref 4.) Menezes, L. F., Lin, C.-C., Zhou, F. & Germino, G. G. Fatty Acid Oxidation is Impaired in An Orthologous Mouse Model of Autosomal Dominant Polycystic Kidney Disease. *EBioMedicine* 5, 183–192 (2016)..

Yes, correct.

They subjected rats with cystic kidney disease (PCK) to the same high lysine diet and found, as expected, no effect of lysine on bodyweight and cyst formation. In addition, diuresis did not change in early or late stages of the disease (the reviewer Figure 3).

Finally, authors challenged humans in a pilot trial with a large peroral load of lysine, followed by 12 hrs urine collection.

Humans with reduced nephron mass (risk for hypertension) have alterations similar to the metabolic alterations observed in rats with kidney damage and hypertension. In addition, the data revealed similar metabolic capacities, including increased lysine modification and reduced TCA excretion, a pattern similar to the protective effect in lysine rats (new Figure 8).

Yes, correct.

4) While the lysine treatment increased albuminuria in Fig. 2, high lysine in the urine was negatively correlated with proteinuria in Fig. 4. Is there any difference between the two conditions?

Authors explained that in acute phase, lysine binds to Megalin-Cubilin, prevents albumin reabsorption in PT and releases an excessively filtered albumin accumulated in the kidney (Figure 4). Induction of albuminuria can be seen only at very early time points and in longer lysine administered time period, lysine improves kidney metabolism by ameliorating injury and reducing fibrosis (Figure 3,5,6).

Yes, correct.

5) Previous reports already showed ketogenic diet could ameliorate kidney injury with weight loss. Although the ketogenic diet could recreate similar conditions to the lysine treatment, it would not be supportive in the context. A lot of metabolic changes could happen not only in the kidney but other organs with lysine treatment and the ketogenic diet. The metabolic changes may be different between lysine treatment and the ketogenic diet. The data shown in Fig. 6 are not enough to convince readers.

They refocused the manuscript by removing these data (previous manuscript Figure 6) and added further data on a spontaneous hypertensive rat and humans with kidney disease (new Figure 7).

Yes, correct.

Minor issues are properly corrected

Yes, correct. Thank you for your review.

Reviewer #2 (Remarks to the Author):

The authors have addressed all major and minor points of criticism in detail and included significant changes into the revised manuscript, convincingly providing new experimental data to confirm their hypothesis. Furthermore, they addressed potential limitations of their study appropriately.

Thank you for your positive review of the manuscript and our revisions.

Although kidney damage is the main focus, possible changes due to long-term lysine intake in other organs still need a more detailed analysis before studying long-term lysine intake in patients. Rinschen and coworkers have already started first investigations, showing that heart, muscle and liver respond to lysine, but not the brain. This difference should be explained in appropriate detail in

future experiments.

We agree with this statement.